# Locally Adaptive Federated Learning

## Abstract

Federated learning is a paradigm of distributed machine learning in which multiple clients coordinate with a central server to learn a model, without sharing their own training data. Standard federated optimization methods such as Federated Averaging (FedAvg) ensure balance among the clients by using the same stepsize for local updates on all clients. However, this means that all clients need to respect the global geometry of the function which could yield slow convergence. In this work, we propose locally adaptive federated learning algorithms, that leverage the local geometric information for each client function. We show that such locally adaptive methods with uncoordinated stepsizes across all clients can be particularly efficient in interpolated (overparameterized) settings, and analyze their convergence in the presence of heterogeneous data for convex and strongly convex settings. We validate our theoretical claims by performing illustrative experiments for both i.i.d. non-i.i.d. cases. Our proposed algorithms match the optimization performance of tuned FedAvg in the convex setting, outperform FedAvg as well as state-of-the-art adaptive federated algorithms like FedAMS for non-convex experiments, and come with superior generalization performance.

## 1 Introduction

Federated Learning (FL) (Kairouz et al., 2021) has become popular as a collaborative learning paradigm where multiple clients jointly train a machine learning model without sharing their local data. Despite the recent success of FL, state-of-the-art federated optimization methods like FedAvg (McMahan et al., 2017) still face various challenges in practical scenarios such as not being able to adapt according to the training dynamics—FedAvg using vanilla SGD updates with constant stepsizes maybe unsuitable for heavy-tail stochastic gradient noise distributions, arising frequently in training large-scale models such as ViT (Dosovitskiy et al., 2021). Such settings benefit from adaptive stepsizes, which use some optimization statistics (e.g., loss history, gradient norm).

In the centralized setting, adaptive methods such as Adam (Kingma & Ba, 2014) and AdaGrad (Duchi et al., 2011) have succeeded in obtaining superior empirical performance over SGD for various machine learning tasks. However, extending adaptive methods to the federated setting remains a challenging task, and majority of the recently proposed adaptive federated methods such as FedAdam (Reddi et al., 2021) and FedAMS (Wang et al., 2022a) consider only server-side adaptivity, i.e., essentially adaptivity only in the aggregation step. Some methods like Local-AMSGrad (Chen et al., 2020) and Local-AdaAlter (Xie et al., 2019) do consider local (client-side) adaptivity, but they perform some form of stepsize aggregation in the communication round, thereby using the same stepsize on all clients.

Using the same stepsize for all clients, that needs to respect the geometry of the global function, can yield sub-optimal convergence. To harness the full power of adaptivity for federated optimization, we argue that it makes sense to use fully locally adaptive stepsizes on each client to capture the local geometric information of each objective function (Wang et al., 2021), thereby leading to faster convergence. However, such a change is non trivial, as federated optimization with uncoordinated stepsizes on different clients might not convergence. The analysis of locally adaptive methods necessitates developing new proof techniques extending the existing error-feedback framework (Stich, 2018) for federated optimization algorithms (that originally works for equal stepsizes) to work for fully un-coordinated local (client) stepsizes. In this work, we provide affirmative answers to the following open questions:

*(a) Can local adaptivity for federated optimization be useful (faster convergence)? (b) Can we design such a locally adaptive federated optimization algorithm that provably converges?*

To answer (a), we shall see a concrete case in Example 1 along with an illustration in Figure 1, showing locally adaptive stepsizes can substantially speed up convergence. For designing a fully locally adaptive method for federated optimization, we need an adaptive stepsize that would be optimal for each client function. Inspired by the Polyak stepsize (Polyak, 1987) which is designed for gradient descent on convex functions, Loizou et al. (2021) recently proposed the stochastic Polyak step-size (SPS) for SGD. SPS comes with strong convergence guarantees, and needs less tuning compared to other adaptive methods like Adam and AdaGrad. We propose the FedSPS algorithm by incorporating the SPS stepsize in the local client updates. We obtain exact convergence of our locally adaptive FedSPS when interpolation condition is satisfied (overparameterized case common in deep learning problems), and convergence to a neighbourhood for the general case. Reminiscing the fact that Li et al. (2020) showed FedAvg needs decaying stepsizes to converge under heterogeneity, we extend our method to a decreasing stepsize version FedDecSPS (following ideas from DecSPS (Orvieto et al., 2022)), that provides exact convergence in practice, for the general non-interpolating setting without the aforementioned small stepsize assumption. Finally, we experimentally observe that the optimization performance of FedSPS is always on par or better than that of tuned FedAvg and FedAMS, and FedDecSPS is particularly efficient in non-interpolating settings.

**Contributions.** We summarize our contributions as follows:

- We show that local adaptivity can lead to substantially faster convergence. We design the first *fully locally adaptive* method for federated learning called FedSPS, and prove sublinear and linear convergence to the optimum, for convex and strongly convex cases, respectively, under interpolation (Theorem 3). This is in contrast to existing adaptive federated methods such as FedAdam and FedAMS, both of which employ adaptivity only for server aggregation.

- For real-world FL scenarios (such as when the interpolation condition is not satisfied due to client heterogeneity), we propose a practically motivated algorithm FedDecSPS that enjoys local adaptivity and exact convergence also in the non-interpolating regime due to decreasing stepsizes.

- We empirically verify our theoretical claims by performing relevant illustrative experiments to show that our method requires less tuning compared to state-of-the-art algorithms which need extensive grid search. We also obtain competitive performance (both optimization as well as generalization) of the proposed FedSPS and FedDecSPS compared to tuned FedAvg and FedAMS for the convex and non-convex cases in i.i.d. as well as non-i.i.d. settings.

## 1.1 ADDITIONAL RELATED WORK

**Adaptive gradient methods and SPS.** Recently, adaptive stepsize methods that use some optimization statistics have become popular for deep learning applications. Such methods, including Adam (Kingma & Ba, 2014) and AdaGrad (Duchi et al., 2011), work well in practice, but their convergence guarantees depend sometimes on unrealistic assumptions (Duchi et al., 2011). An adaptive method with sound theoretical guarantees is the Polyak stepsize (Polyak, 1987), which has been recently extended to the stochastic setting by Loizou et al. (2021) and termed stochastic Polyak stepsize (SPS).Extensions of SPS have been proposed for solving structured non-convex problems (Gower et al., 2021a) and in the update rule of stochastic mirror descent (D'Orazio et al., 2021). Further follow-up works have come up with various ways to overcome the limitations of vanilla SPS, such as when optimal stochastic loss values are not known (Orvieto et al., 2022; Gower et al., 2022), or when the interpolation condition does not hold (Orvieto et al., 2022; Gower et al., 2021b), as well as a proximal variant for tackling regularization terms (Schaipp et al., 2023).

**Adaptive federated optimization.** Reddi et al. (2021) provide a general framework for adaptive federated optimization (FedOpt), including particular instances such as FedAdam and FedYogi, by using the corresponding centralized adaptive methods as the server optimizer. Several works followed on the idea of server side adaptivity, some recent ones being CD-Adam (Wang et al., 2022b) and FedAMS (Wang et al., 2022a). Fully locally adaptive stepsizes on the client side have not been explored before, except in one concurrent work Kim et al. (2023). Their proposed method is based on estimator for the inverse local Lipschitz constant from (Malitsky & Mishchenko, 2019), and analyses only the non-convex setting a strong bounded gradient assumption.

## 2 PROBLEM SETUP

In this work, we consider the following sum-structured (cross-silo) federated optimization problem

$$f^\star := \min_{\mathbf{x} \in \mathbb{R}^d} \left[ f(\mathbf{x}) := \frac{1}{n} \sum_{i=1}^n f_i(\mathbf{x}) \right], \tag{1}$$

where the components $f_i \colon \mathbb{R}^d \to \mathbb{R}$ are distributed among $n$ local clients and are given in stochastic form $f_i(\mathbf{x}) := \mathbb{E}_{\xi \sim \mathcal{D}_i}[F_i(\mathbf{x}, \xi)]$, where $\mathcal{D}_i$ denotes the distribution of $\xi$ over parameter space $\Omega_i$ on client $i \in [n]$. Standard empirical risk minimization is an important special case of this problem, when each $\mathcal{D}_i$ presents a finite number $m_i$ of elements $\{\xi_1^i, \dots, \xi_{m_i}^i\}$. Then $f_i$ can be rewritten as $f_i(\mathbf{x}) = \frac{1}{m_i} \sum_{j=1}^{m_i} F_i(\mathbf{x}, \xi_j^i)$. We do not make any restrictive assumptions on the data distributions $\mathcal{D}_i$, so our analysis covers the case of heterogeneous (non-i.i.d.) data where $\mathcal{D}_i \neq \mathcal{D}_j, \forall i \neq j$ and the *local minima* $\mathbf{x}_i^\star := \arg\min_{\mathbf{x} \in \mathbb{R}^d} f_i(\mathbf{x})$, can be different from the *global minimizer* of (1).

## 3 LOCALLY ADAPTIVE FEDERATED OPTIMIZATION

In the following, we provide a background on federated optimization, and the (stochastic) Polyak stepsize. This is followed by an Example to motivate how local adaptivity with (stochastic) Polyak stepsizes can help to improve convergence—especially in the interpolation regime. Finally, we outline our proposed method FedSPS to solve (1).

### 3.1 BACKGROUND AND MOTIVATION

**Federated averaging.** A common approach to solving (1) in the distributed setting is FedAvg (McMahan et al., 2017) also known as Local SGD (Stich, 2018). This involves the clients performing a local step of SGD in each iteration, and the clients communicate with a central server after every $\tau$ iterations—their iterates are averaged on the server, and sent back to all clients. FedAvg corresponds to the special case of Algorithm 1 with constant stepsizes $\gamma_t^i \equiv \gamma_0$ (Line 4).

**PS and SPS.** Considering the centralized setting ($n = 1$) of finite-sum optimization on a single worker $\min_{\mathbf{x} \in \mathbb{R}^d} \left[ f_1(\mathbf{x}) := \frac{1}{m} \sum_{j=1}^m F_1(\mathbf{x}, \xi_j^1) \right]$, we introduce the PS as well as the SPS as below:

- **Deterministic Polyak stepsize.** The convergence analysis of Gradient Descent (GD) for a convex function $f_1(\mathbf{x})$ involves the inequality $\|\mathbf{x}_{t+1} - \mathbf{x}^\star\|^2 \leq \|\mathbf{x}_t - \mathbf{x}^\star\|^2 - 2\gamma_t (f_1(\mathbf{x}_t) - f_1(\mathbf{x}^\star)) + \gamma_t^2 \|\nabla f_1(\mathbf{x}_t)\|^2$, the right-hand side of which is minimized by the PS $\gamma_t = \frac{f_1(\mathbf{x}_t) - f_1^\star}{\|\nabla f_1(\mathbf{x}_t)\|^2}$.

- **Stochastic Polyak stepsize.** We use the notion of SPS$_{\max}$ from the original paper (Loizou et al., 2021). The SPS$_{\max}$ stepsize for SGD (with single stochastic sample) is given by $\gamma_t = \min\left\{ \frac{F_1(\mathbf{x}_t, \xi_j^1) - F_1^\star}{c\|\nabla F_1(\mathbf{x}_t, \xi_j^1)\|^2}, \gamma_b \right\}$, where $F_1^\star := \inf_{\xi \in \mathcal{D}_1, \mathbf{x} \in \mathbb{R}^d} F_1(\mathbf{x}, \xi)$, $\gamma_b > 0$ is an upper bound on the stepsize that controls the size of neighbourhood ($\gamma_b$ trades-off adaptivity for accuracy), and $c > 0$ is a constant scaling factor. Instead of using the optimal function values of each stochastic function as in the original paper, we use the lower bound on the function values $\ell_1^\star \leq F_1^\star$, which is easier to obtain for many practical tasks as shown in (Orvieto et al., 2022).

**Example 1** (Local adaptivity using Polyak stepsizes can improve convergence). *For a parameter $a > 0$, consider the finite sum optimization problem $\min_{x \in \mathbb{R}} \left[ f(x) := \frac{1}{2} \sum_{i=1}^2 f_i(x) \right]$, with $f_1(x) = \frac{a}{2}x^2$, $f_2(x) = \frac{1}{2}x^2$ in the interpolation regime. If we solve this problem using mini-batch GD, $x_{t+1} = x_t - \frac{\gamma}{2}(\nabla f_1(x_t) + \nabla f_2(x_t))$, we are required to choose a stepsize $\gamma \leq 2/L$, where $L = \frac{1+a}{2}$ to enable convergence, and therefore $\Omega(a \log \frac{1}{\epsilon})$ steps are needed. However, if we solve the same problem using locally adaptive distributed GD of the form $x_{t+1} = x_t - \frac{1}{2}(\gamma_1 \nabla f_1(x_t) + \gamma_2 \nabla f_2(x_t))$, then the complexity can be near-constant. Concretely, for any stepsizes $\gamma_i \in [\frac{1}{2}\gamma_i^\star, \frac{3}{2}\gamma_i^\star]$, with $\gamma_1^\star = \frac{1}{a}$, $\gamma_2^\star = 1$ (which also includes the Polyak stepsizes corresponding to functions $f_1$ and $f_2$), the iteration complexity is $\mathcal{O}(\log \frac{1}{2})$, which can be arbitrarily better than $\Omega(a \log \frac{1}{\epsilon})$ when $a \to \infty$. Note that the observation made in this example can also be extended to the stochastic case of SGD with SPS, as illustrated in Figure 1.*

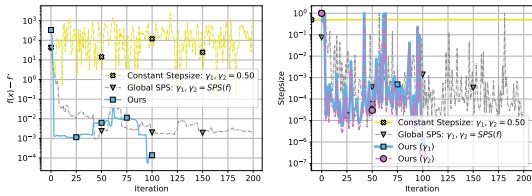

Figure 1: Illustration for Example 1, showing local adaptivity can improve convergence. We run SGD with constant, global SPS, and locally adaptive SPS stepsizes (with $c = 0.5, \gamma_b = 1.0$), for functions $f_1(x) = x^2$, $f_2(x) = \frac{1}{2}x^2$, where stochastic noise was simulated by adding Gaussian noise with mean 0, and standard deviation 10 to the gradients.

## 3.2 Proposed Method

Motivated by the previous example on the benefit of local adaptivity, we now turn to design such a locally adaptive federated optimization algorithm with provable convergence guarantees. As stated before, we need an adaptive stepsize that is optimal for each client function, and we choose the SPS stepsize for this purpose. In the following, we describe our proposed method FedSPS.

**FedSPS.** We propose a fully locally (i.e., client-side) adaptive federated optimization algorithm FedSPS (Algorithm 1) with asynchronous stepsizes, i.e., the stepsizes are different across the clients, and also across the local steps for a particular client. The FedSPS stepsize for a client $i$ and local iteration $t$ will be given by

$$\gamma_t^i = \min \left\{ \frac{F_i(\mathbf{x}_t^i, \xi_t^i) - \ell_i^\star}{c \left\| \nabla F_i(\mathbf{x}_t^i, \xi_t^i) \right\|^2}, \gamma_b \right\} , \tag{2}$$

where $c$, $\gamma_b > 0$ are constants as explained before, $\xi_t^i$ is the sample at time $t$ on worker $i$, $F_i(\mathbf{x}_t^i, \xi_t^i)$ is the stochastic loss, $\mathbf{g}_t^i := \nabla F_i(\mathbf{x}_t^i, \xi_t^i)$ is the stochastic gradient, and $\ell_i^\star \leq F_i^\star = \inf_{\xi^i \in \mathcal{D}_i, \mathbf{x} \in \mathbb{R}^d} F_i(\mathbf{x}, \xi^i)$ is a lower bound on the minima of all functions on worker $i$. Since the loss functions are non-negative for most practical machine learning tasks, we can use $\ell_i^\star = 0$ as discussed before, for running our algorithms. We analyse FedSPS in the strongly convex and convex settings and prove convergence guarantees (Theorem 3). We would like to stress that $\gamma_b$ and $c$ are free hyperparameters (in the sense that they do not depend theoretically on any problem-dependent parameters) and we demonstrate that they require minimal tuning through relevant sensitivity analysis in Section 6. This is an advantage over the learning rate parameter of FedAvg and FedAMS which theoretically depend on $L$, and require extensive grid search in practice (Section F.4). The notations used throughout can be extended to the mini-batch setting as described in Appendix C.1.

**Remark 2** (Alternative design choices)**.** *Note that there can be various alternative design choices for incorporating SPS in FedAvg. We tried some variants such as FedSPS-Normalized using client and server side correction to account for solution bias (Wang et al., 2021) due to asynchronous stepsizes (Appendix D.1). We also introduce FedSPS-Global in the Appendix D.2, that uses aggregation of stepsizes in communication rounds, similar to (Chen et al., 2020; Xie et al., 2019). However, none of these other design choices provided any practical advantages over our proposed FedSPS.*

## 4 Convergence analysis of FedSPS

### 4.1 Assumptions on the objective function and noise

**Assumption 1** (*L*-smoothness)**.** *Each function $F_i(\mathbf{x}, \xi) \colon \mathbb{R}^d \times \Omega_i \to \mathbb{R}$, $i \in [n]$ is differentiable for each $\xi \in \mathrm{supp}(\mathcal{D}_i)$ and there exists a constant $L \geq 0$ such that for each $\mathbf{x}, \mathbf{y} \in \mathbb{R}^d, \xi \in \mathrm{supp}(\mathcal{D}_i)$:*

$$\left\| \nabla F_i(\mathbf{y}, \xi) - \nabla F_i(\mathbf{x}, \xi) \right\| \leq L \left\| \mathbf{x} - \mathbf{y} \right\| . \tag{3}$$

Note that Assumption 1 implies $L$-smoothness of each $f_i(\mathbf{x})$ and of $f(\mathbf{x})$. The assumption of each $F_i(\mathbf{x}, \xi)$ being smooth is often used in federated and decentralized optimization literature (for e.g., (Koloskova et al., 2020, Assumption 1a), or (Koloskova et al., 2022, Assumption 3)).

**Algorithm 1 FedSPS:** Federated averaging with fully locally adaptive stepsizes.

---

**Input:** $\mathbf{x}_0^i = \mathbf{x}_0, \forall i \in [n]$
1: **for** $t = 0, 1, \cdots, T-1$ **do**
2:     **for** each client $i = 1, \cdots, n$ in parallel **do**
3:         sample $\xi_t^i$, compute $\mathbf{g}_t^i := \nabla F_i(\mathbf{x}_t^i, \xi_t^i)$
4:         **FedSPS:** $\gamma_t^i = \min\left\{ \frac{F_i(\mathbf{x}_t^i, \xi_t^i) - \ell_i^\star}{c\|\mathbf{g}_t^i\|^2}, \gamma_b \right\}$             ▷ local stochastic Polyak stepsize
5:         **if** $t+1$ is a multiple of $\tau$ **then**
6:             $\mathbf{x}_{t+1}^i = \frac{1}{n} \sum_{i=1}^n \left(\mathbf{x}_t^i - \gamma_t^i \mathbf{g}_t^i\right)$            ▷ communication round
7:         **else**
8:             $\mathbf{x}_{t+1}^i = \mathbf{x}_t^i - \gamma_t^i \mathbf{g}_t^i$                     ▷ local step
9:         **end if**
10:     **end for**
11: **end for**

---

**Assumption 2** ($\mu$-convexity). *There exists a constant $\mu \geq 0$ such that for each for each $i \in [n]$, $\xi \in \mathrm{supp}(\mathcal{D}_i)$ and for all $\mathbf{x}, \mathbf{y} \in \mathbb{R}^d$:*

$$F_i(\mathbf{y}, \xi) \geq F_i(\mathbf{x}, \xi) + \langle \nabla F_i(\mathbf{x}, \xi), \mathbf{y} - \mathbf{x} \rangle + \frac{\mu}{2} \|\mathbf{y} - \mathbf{x}\|^2 . \tag{4}$$

For some of our results, we assume $\mu$-strong convexity for a parameter $\mu > 0$, or convexity (when $\mu = 0$). Furthermore we assume (as mentioned in the introduction) access to stochastic functions $F_i(\mathbf{x}, \xi)$ on each client $i$, with $E_{\xi \sim \mathcal{D}_i} \nabla F_i(\mathbf{x}, \xi) = \nabla f_i(\mathbf{x})$, $E_{\xi \sim \mathcal{D}_i} F_i(\mathbf{x}, \xi) = f_i(\mathbf{x})$.

**Finite optimal objective difference.** For each $i \in [n]$ we denote $f_i^\star := \inf_{\mathbf{x} \in \mathbb{R}^d} f_i(\mathbf{x})$. Recall that we defined $F_i^\star := \inf_{\xi \in \mathcal{D}_i, \mathbf{x} \in \mathbb{R}^d} F_i(\mathbf{x}, \xi)$, and need knowledge of lower bounds, $\ell_i^\star \leq F_i^\star$ for our algorithm. We define the quantity

$$\sigma_f^2 := \frac{1}{n} \sum_{i=1}^n \left(f_i(\mathbf{x}^\star) - \ell_i^\star\right) = f^\star - \frac{1}{n} \sum_{i=1}^n \ell_i^\star , \tag{5}$$

that will appear in our complexity estimates, and thus we implicitly assume that $\sigma_f^2 < \infty$ (finite optimal objective difference). Moreover, $\sigma_f^2$ also acts as our measure of heterogeneity between clients. This is in line with previous works on federated optimization in non-i.i.d. setting, such as Li et al. (2020) that used $\Gamma := f^\star - E_i f_i^\star$ as the heterogeneity measure. We can relate $\sigma_f^2$ to the more standard measures of function heterogeneity $\zeta^2 = \frac{1}{n} \sum_{i=1}^n \|\nabla f_i(\mathbf{x}) - \nabla f(\mathbf{x})\|_2^2$ and gradient variance $\sigma^2 = \frac{1}{n} \sum_{i=1}^n \mathbb{E}_{\xi^i} \|\nabla F_i(\mathbf{x}, \xi^i) - \nabla f_i(\mathbf{x})\|^2$ in the federated literature (Koloskova et al., 2022; Wang et al., 2022a) as shown in the following proposition (proof in Appendix C.2). For the case of convex functions, it suffices (Koloskova et al., 2020) to compare with $\zeta_\star^2 = \frac{1}{n} \sum_{i=1}^n \|\nabla f_i(\mathbf{x}^\star)\|_2^2$, $\sigma_\star^2 = \frac{1}{n} \sum_{i=1}^n \mathbb{E}_{\xi^i} \|\nabla F_i(\mathbf{x}^\star, \xi^i) - \nabla f_i(\mathbf{x}^\star)\|^2$, calculated at the global optimum $\mathbf{x}^\star = \arg\min_{\mathbf{x} \in \mathbb{R}^d} f(\mathbf{x})$. We can observe that $\sigma_f^2$ is actually a stronger assumption than bounded noise at optimum ($\zeta_\star, \sigma_\star$), but weaker than uniformly bounded noise ($\zeta, \sigma$).

**Proposition 1** (Comparison of heterogeneity measures). *Using the definitions of $\sigma_f^2$, $\zeta_\star^2$, and $\sigma_\star^2$ as defined above, we have: (a) $\zeta_\star^2 \leq 2L\sigma_f^2$, and (b) $\sigma_\star^2 \leq 2L\sigma_f^2$.*

### 4.2 CONVERGENCE OF FULLY LOCALLY ADAPTIVE FEDSPS

In this section we provide the convergence guarantees of FedSPS on sums of convex (or strongly convex) functions. We do not make any restriction on $\gamma_b$, and thus denote this as the fully locally adaptive setting that is of most interest to us. The primary theoretical challenge is extending the error-feedback framework (that originally works for equal stepsizes) (Mania et al., 2017; Stich & Karimireddy, 2020) to work for fully un-coordinated local stepsizes, and we do this for the first time in our work. All proofs are provided in Appendix B.

**Theorem 3** (Convergence of FedSPS). *Assume that Assumptions 1 and 2 hold and $c \geq 2\tau^2$, then after $T$ iterations ($T/\tau$ communication rounds) of FedSPS (Algorithm 1) it holds*

*(a) Convex case:*

$$\frac{1}{Tn} \sum_{t=0}^{T-1} \sum_{i=1}^{n} \mathbb{E}[f_i(\mathbf{x}_t^i) - \ell_i^{\star}] \leq \frac{2}{T\alpha} \left\| \bar{\mathbf{x}}_0 - \mathbf{x}^{\star} \right\|^2 + \frac{4\gamma_b \sigma_f^2}{\alpha}, \tag{6}$$

*where $\alpha := \min\left\{\frac{1}{2cL}, \gamma_b\right\}$. If $\mu > 0$, and $c \geq 4\tau^2$, we have*
*(b) Strongly convex case:*

$$\mathbb{E} \left\| \bar{\mathbf{x}}_T - \mathbf{x}^{\star} \right\|^2 \leq A(1 - \mu\alpha)^T \left\| \bar{\mathbf{x}}_0 - \mathbf{x}^{\star} \right\|^2 + \frac{2\gamma_b \sigma_f^2}{\alpha\mu}, \tag{7}$$

*where $A = \frac{1}{\mu\alpha}$, and $\bar{\mathbf{x}}_t := \frac{1}{n} \sum_{i=1}^{n} \mathbf{x}_t^i$.*

The convergence criterion of the first result (6) is non-standard, as it involves the average of all iterates $\mathbf{x}_t^i$ on the left hand side, and not the average $\bar{\mathbf{x}}_t$ more commonly used. However, note that every $\tau$-th iteration these quantities are the same, and thus our result implies convergence of $\frac{1}{Tn} \sum_{t=0}^{(T-1)/\tau} \sum_{i=1}^{n} \mathbb{E}[f_i(\bar{\mathbf{x}}_{t\tau}) - \ell_i^{\star}]$. Moreover, in the interpolation case all $\ell_i^{\star} \equiv f^{\star}$.

**Remark 4** (Minimal need for hyperparamter tuning). *The parameter $\tau$ is a user selected input parameter to determine the number of local steps, and $\gamma_b$ trades-off adaptivity (potentially faster convergence for large $\gamma_b$) and accuracy (higher for small $\gamma_b$). Moreover, as $c$ only depends on the input parameter $\tau$ and not on properties of the function (e.g. $L$ or $\mu$), it is also a free parameter. The algorithm provably converges (up to the indicated accuracy) for any choice of these parameters. The lower bounds $\ell_i^{\star}$ can be set to zero for many machine learning problems as discussed before. Therefore, we effectively reduce the need for hyperparameter tuning compared to previous methods like FedAMS whose convergence depended on problem-dependent parameters.*

**Comparison with SPS (Loizou et al., 2021).** We will now compare our results to Loizou et al. (2021) that studied SPS for a single worker ($n = 1$). First, we note that in the strongly convex case, we almost recover Theorem 3.1 in Loizou et al. (2021). The only differences are that they have $A = 1$ and allow weaker bounds on the parameter $c$ ($c \geq 1/2$, vs. our $c > 4$), but we match other constants. In the convex case, we again recover (Loizou et al., 2021, Theorem 3.4) up to constants and the stronger condition on $c$ (vs. $c > 1$ in their case).

• **(Special case I) Exact convergence of FedSPS in interpolation regime:** We highlight the linear convergence of FedSPS in the interpolation case ($\sigma_f = 0$) in the following corollary.

**Corollary 5** (Linear Convergence of FedSPS under Interpolation). *Assume interpolation, $\sigma_f^2 = 0$ and let the assumptions of Theorem 3 be satisfied with $\mu > 0$. Then*

$$\mathbb{E} \left\| \bar{\mathbf{x}}_T - \mathbf{x}^{\star} \right\|^2 \leq A(1 - \mu\alpha)^T \left\| \bar{\mathbf{x}}_0 - \mathbf{x}^{\star} \right\|^2. \tag{8}$$

• **(Special case II) Exact convergence of FedSPS in the small stepsize regime:** Theorem 3 shows convergence of FedSPS to a neighborhood of the solution. Decreasing $\gamma_b$ (smaller than $\frac{1}{2cL}$) can improve the accuracy, but the error is at least $\Omega(\sigma_f^2)$ even when $\gamma_b \to 0$. This issue is also persistent in the original work on $\text{SPS}_{\max}$ (Loizou et al., 2021, Corr. 3.3). However, we remark when the stepsize upper bound $\gamma_b$ is chosen extremely small—not allowing for adaptivity—-then FedSPS becomes identical to constant stepsize FedAvg. This is not reflected in Theorem 3 that cannot recover the exact convergence known for FedAvg. We address this in the next theorem, proving exact convergence of small stepsize FedSPS (equivalent to analysis of FedAvg with $\sigma_f^2$ assumption).

**Theorem 6** (Convergence of small stepsize FedSPS). *Assume that Assumptions 1 and 2 hold and $\gamma_b \leq \min\left\{\frac{1}{2cL}, \frac{1}{20L\tau}\right\}$, then after $T$ iterations of FedSPS (Algorithm 1) it holds*
*(a) Convex case:*

$$\frac{1}{T} \sum_{t=0}^{T-1} \mathbb{E}[f(\bar{\mathbf{x}}_t) - f^{\star}] = \mathcal{O}\left( \frac{1}{T\gamma_b} \left\| \bar{\mathbf{x}}_0 - \mathbf{x}^{\star} \right\|^2 + \gamma_b L \sigma_f^2 + \gamma_b^2 L \tau^2 \sigma_f^2 \right), \tag{9}$$

*and when $\mu > 0$,*
*(b) Strongly convex case:*

$$\mathbb{E} \left\| \bar{\mathbf{x}}_T - \mathbf{x}^{\star} \right\|^2 = \mathcal{O}\left( \frac{\left\| \bar{\mathbf{x}}_0 - \mathbf{x}^{\star} \right\|^2}{\mu\gamma_b} (1 - \mu\gamma_b)^T + \gamma_b \frac{L\sigma_f^2}{\mu} + \gamma_b^2 \frac{L\tau^2 \sigma_f^2}{\mu} \right). \tag{10}$$

This theorem shows that by choosing an appropriately small $\gamma_b$, any arbitrary target accuracy $\epsilon > 0$ can be obtained. We are only aware of Li et al. (2020) that studies FedAvg under similar assumptions as us ($\Gamma := f^\star - E_i f_i^\star$ measuring heterogeneity). However, their analysis additionally required bounded stochastic gradients and their convergence rates are weaker (e.g., not recovering linear convergence under interpolation when $\sigma_f^2 = 0$).

## 5 DECREASING FEDSPS FOR EXACT CONVERGENCE

In the previous section, we have proved that FedSPS converges in the interpolation setting irrespective of the value of the stepsize parameter $\gamma_b$. However, many practical federated learning scenarios such as those involving heterogeneous clients constitute the non-interpolating setting ($\sigma_f > 0$). Here, we need to choose a small value of $\gamma_b$ to ensure convergence, trading-off adaptivity for achieving exact convergence. We might recall that Li et al. (2020) proved choosing a decaying stepsize is necessary for convergence of FedAvg under heterogeneity. In this section, we draw inspiration from the decreasing SPS stepsize DecSPS (Orvieto et al., 2022), to develop FedDecSPS, that achieves exact convergence in practical non-interpolating scenarios without compromising adaptivity.

**FedDecSPS.** In order to obtain exact convergence to arbitrary accuracy (without the small stepsize assumption) in the heterogeneous setting with $\sigma_f > 0$, we propose a heuristic decreasing stepsize version of FedSPS, called FedDecSPS. The FedDecSPS stepsize for client $i$ and local iteration $t$ is given by $\gamma_t^i = \frac{1}{c_t} \min \left\{ \frac{F_i(\mathbf{x}_t^i, \xi_t^i) - \ell_i^\star}{\left\| \nabla F_i(\mathbf{x}_t^i, \xi_t^i) \right\|^2}, c_{t-1} \gamma_{t-1}^i \right\}$, where $(c_t)_{t=0}^\infty$ is any non-decreasing positive sequence of real numbers. We also set $c_{-1} = c_0$, $\gamma_{-1}^i = \gamma_b$. Experiments involving heterogeneous clients in Section 6 demonstrate the practical convergence benefits of FedDecSPS in non-interpolating settings.

## 6 EXPERIMENTS

**Experimental setup.** For all federated training experiments we have 500 communication rounds (the no. of communication rounds being $T/\tau$ as per our notation), 5 local steps on each client ($\tau = 5$, unless otherwise specified for some ablation experiments), and a batch size of 20 ($|\mathcal{B}| = 20$). We perform experiments in the i.i.d. as well as non-i.i.d. settings. Results are reported for both settings without client sampling (10 clients) and with client sampling (10 clients sampled uniformly at random from 100 clients with participation fraction 0.1, and data split among all 100 clients) i.e., $n = 10$ active clients throughout. The i.i.d. experiments involve randomly shuffling the data and equally splitting the data between clients. For non-i.i.d. experiments, we assign every client samples from exactly two classes of the dataset, the splits being non-overlapping and balanced with each client having same number of samples (Li et al., 2020). Our code is based on publicly available repositories for SPS and FedAMS[1], and will be made available upon acceptance.

**FedSPS.** The implementation is done according to Algorithm 1. Since all our experimental settings involve non-negative loss functions, we can use the lower bound $\ell_i^\star = 0$ (Orvieto et al., 2022), throughout. In the following, we perform empirical sensitivity analysis for the free hyperparameters $\gamma_b$ and $c$, concluding that our method is indeed insensitive to changes in these parameters.

We start with benchmarking our method by running some initial convex experiments performing classification of the MNIST (i.i.d.) dataset (LeCun et al., 2010) with a logistic regression model, without client sampling. In Figure 2(a), we compare the effect of varying $\gamma_b \in \{1, 5, 10\}$ on FedSPS, and varying $\gamma \in \{0.1, 0.01\}$ on FedAvg. We find that FedAvg is not robust to changing stepsize—converging well for stepsize 0.1, but very slow convergence for stepsize 0.01. On the contrary, all instances of FedSPS converge to a neighbourhood of the optimum—the size of the neighbourhood being proportional to $\gamma_b$ as suggested by the theory. We now fix $\gamma_b = 1$, and perform an ablation study to understand the effect of varying SPS scaling parameter $c$ on the convergence in Figure 2(c). For number of local steps $\tau = 5$, we vary $c$ from 0.01 to 40 (i.e., of the order of square of $\tau$). Unlike what is predicted by our theory, empirically we observe that small $c$ works better and larger $c$ leads to slower convergence. Moreover, all values of $c \in \{0.01, 0.1, 0.5, 1.0\}$ have similarly good

---

[1]SPS (https://github.com/IssamLaradji/sps), FedAMS (https://github.com/jinghuichen/FedCAMS)

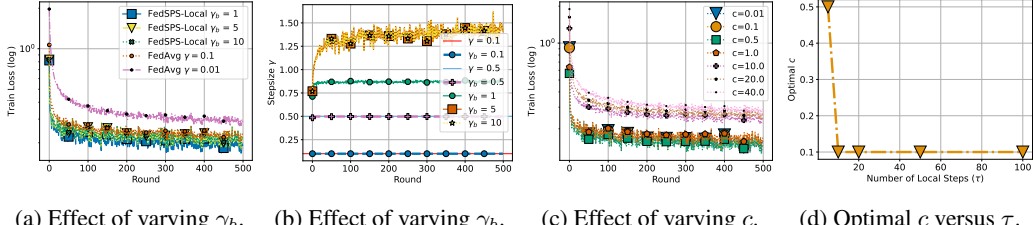

(a) Effect of varying $\gamma_b$.  (b) Effect of varying $\gamma_b$.  (c) Effect of varying $c$.  (d) Optimal $c$ versus $\tau$.

Figure 2: Sensitivity analysis of FedSPS to hyperparameters for convex logistic regression on the MNIST dataset (i.i.d.) without client sampling. (a) Comparing the effect of varying $\gamma_b$ on FedSPS and varying $\gamma$ on FedAvg convergence—FedAvg is more sensitive to changes in $\gamma$, while FedSPS is insensitive changes in to $\gamma_b$. (b) Effect of varying $\gamma_b$ on FedSPS stepsize adaptivity—adaptivity is lost if $\gamma_b$ is chosen too small. (c) Small $c$ works well in practice ($\tau = 5$). (d) Optimal $c$ versus $\tau$, showing that there is no dependence.

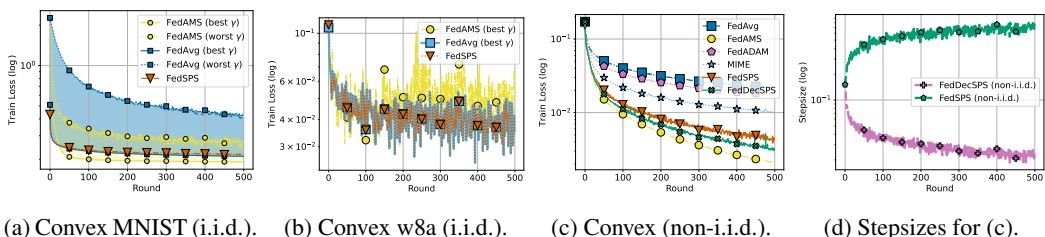

(a) Convex MNIST (i.i.d.).  (b) Convex w8a (i.i.d.).  (c) Convex (non-i.i.d.).  (d) Stepsizes for (c).

Figure 3: Comparison for convex logistic regression. (a) MNIST dataset (i.i.d. without client sampling). (b) w8a dataset (i.i.d. with client sampling). (c) MNIST dataset (non-i.i.d. with client sampling). (d) Average stepsize across all clients for FedSPS and FedDecSPS corresponding to (c). Performance of FedSPS matches that of FedAvg with *best tuned local learning rate* for the i.i.d. cases, and outperforms in the non-i.i.d. case.

convergence, thereby implying our method is robust to this hyperparameter and needs no tuning. We provide additional plots for $\tau \in \{10, 20, 50, 100\}$ local steps in Appendix F.2 to confirm that this observation is valid across all values of $\tau$, and plot the optimal value of $c$ versus $\tau$ for each case in Figure 2(d). Gaining insights from above experiments we fix $c = 0.5$ for all further experiments.

**FedDecSPS.** We evaluate the performance of FedDecSPS with $c_t = c_0\sqrt{t+1}$. Similar to the sensitivity analysis of FedSPS towards $c$ (Figure 2), we performed ablations studies for a fixed value of $\gamma_b$ and varying $c_0$ as well as $\tau$. The observation is same as the previous case: the optimal value of $c_0$ does not scale according to $\tau$ as suggested by theory and we fix $c_0 = 0.5$ for all experiments. Similarly we fix $\gamma_b = 1$, following similar observations as before. We compare the convergence of FedSPS and FedDecSPS for the case of heterogeneous data on clients (i.e., $\sigma_f > 0$) in Figure 3 (c) and (d), as well as Figure 5. We observe that our practically motivated FedDecSPS performs better in such non-interpolating settings, as expected.

**FedAvg and FedAMS.** We compare the performance of our methods—FedSPS and FedDecSPS against the FedAvg baseline, and the state-of-the-art adaptive federated algorithm FedAMS Wang et al. (2022a). FedAvg and FedAMS need extensive tuning using grid search for client learning rate $\eta_l$, server learning rate $\eta$, as well as max stabilization factor $\epsilon$, and $\beta_1$, $\beta_2$. We refer readers to Appendix F.4 for details on the grid search performed and the optimal set of hyperparameters.

**Convex comparison.** For the convex setting of logistic regression on MNIST dataset (i.i.d. setting), without client sampling, we now compare FedSPS with FedAvg and FedAMS in Figure 3(a). We see that the convergence of FedSPS matches that of the *best tuned FedAvg*. Note that while the *best tuned FedAMS* slightly outperforms our method, it requires considerable tuning depicted by the large margin between best and worst learning rate performances. For additional convex experiments in the more practical setting with client sampling, we take the problem of binary classification of LIBSVM Chang & Lin (2011) datasets (w8a, mushrooms, ijcnn, phishing, a9a) with logistic regression model in the i.i.d. setting. We report the performance on w8a in Figure 3(b), where FedSPS again converges similarly as tuned FedAvg, and better than FedAMS. We defer rest of the LIBSVM dataset plots to Appendix F. In the non-i.i.d. case we compare our proposed FedSPS and FedDecSPS to the FedAvg baseline, adaptive federated methods FedAMS and FedADAM, as well as another state-of-the-art

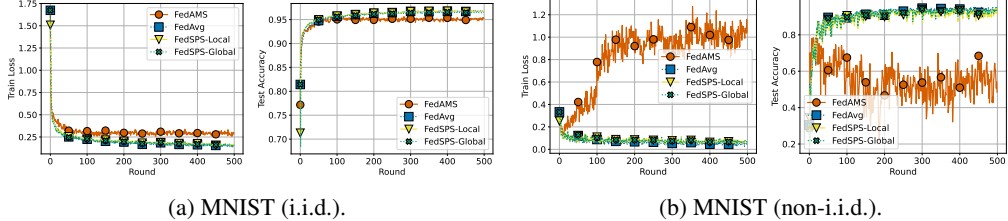

(a) MNIST (i.i.d.).         (b) MNIST (non-i.i.d.).

Figure 4: Non-convex MNIST experiments with client sampling. (a) Non-convex case of LeNet on MNIST dataset (i.i.d.). (b) Non-convex case of LeNet on MNIST dataset (non-i.i.d.). First column represents training loss, second column is test accuracy. Convergence of FedSPS is very close to that of FedAvg with the *best possible tuned local learning rate*. Moreover, FedSPS converges better than FedAMS for the non-convex MNIST case (both i.i.d. and non-i.i.d.), and also offers superior generalization performance than FedAMS. FedSPS is referred to as FedSPS-Local here in the legends, to distinguish it clearly from FedSPS-Global.

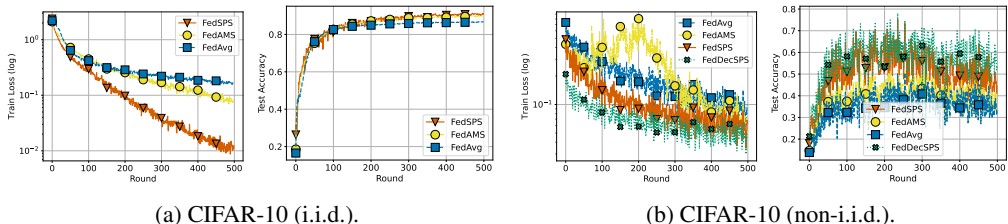

(a) CIFAR-10 (i.i.d.).        (b) CIFAR-10 (non-i.i.d.).

Figure 5: Non-convex CIFAR-10 experiments with client sampling. (a) Non-convex case of ResNet18 on CIFAR-10 dataset (i.i.d.). (b) Non-convex case of ResNet18 on CIFAR-10 dataset (non-i.i.d.). First column represents training loss, second column is test accuracy. FedSPS converges better than FedAvg and FedAMS for both i.i.d. and non-i.i.d. settings, and also offers superior generalization performance.

federated method MIME (Karimireddy et al., 2021). In this setting FedDecSPS does better than FedSPS, and our methods outperform the *best tuned* FedAvg, FedADAM and MIME.

**Non-convex comparison.** For non-convex experiments, we look at multi-class classification of MNIST dataset using LeNet architecture (LeCun et al., 1998) and CIFAR-10 dataset using ResNet18 architecture (He et al., 2016) in the i.i.d. as well as non-i.i.d. setting (with client sampling), in Figures 4 and 5. For the upper bound on stepsizes, we use the *smoothing* technique for rest of the experiments, as suggested by Loizou et al. (2021) for avoiding sudden fluctuations in the stepsize. For a client $i \in [n]$ and iteration $t$, the adaptive iteration-dependent upper bound is given by $\gamma_{b,t}^i = 2^{|\mathcal{B}|/m_i} \gamma_{b,t-1}^i$, where $|\mathcal{B}|$ is the batch-size, $m_i$ is the number of data examples on that client and we fix $\gamma_{b,0} = 1$. In Figure 4 (MNIST), we find that FedSPS and FedSPS-Global converge almost identically, and their convergence is also very close to that of FedAvg with the best possible tuned local learning rate, while outperforming FedAMS. In Figure 5 (CIFAR-10), FedSPS and FedDecSPS outperform tuned FedAvg and FedAMS in terms of both training loss and test accuracy.

# 7 CONCLUSION

In this paper, we show that locally adaptive federated optimization can lead to faster convergence by harnessing the geometric information of local objective functions. This is especially beneficial in the interpolating setting, which arises commonly for overparameterized deep learning problems. We propose a locally adaptive federated optimization algorithm FedSPS, by incorporating the stochastic Polyak stepsize in local steps, and prove sublinear and linear convergence to a neighbourhood for convex and strongly convex cases, respectively. We further extend our method to the decreasing stepsize version FedDecSPS, that enables exact convergence even in practical non-interpolating FL settings without compromising adaptivity. We perform relevant illustrative experiments to show that our proposed method is relatively insensitive to the hyperparameters involved, thereby requiring less tuning compared to other state-of-the-art federated algorithms. Moreover, our methods perform as good or better than tuned FedAvg and FedAMS for convex as well as non-convex experiments in i.i.d. and non-i.i.d. settings.

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

# Appendix

The Appendix is organized as follows. We begin by introducing some general definitions and in-equalities used throughout the proofs, in Section A. Proofs for convergence analysis of FedSPS are provided in Section B—including convex and strongly convex cases. Section C provides some additional theoretical details for FedSPS. Alternative design choices for FedSPS, namely FedSPS-Normalized and FedSPS-Global are described in Section D. We provide some theoretical results for FedSPS in the non-convex setting in Section E. Finally, additional experiments for FedSPS and FedSPS-Global in convex settings are provided in Section F.

CONTENTS OF APPENDIX

## A  TECHNICAL PRELIMINARIES

Let us present some basic definitions and inequalities used in the proofs throughout the appendix.

### A.1  GENERAL DEFINITIONS

**Definition 1** (Convexity). *The function $f : \mathbb{R}^d \to \mathbb{R}$, is convex, if for all $\mathbf{x}, \mathbf{y} \in \mathbb{R}^d$*

$$f(\mathbf{x}) \geq f(\mathbf{y}) + \langle \nabla f(\mathbf{y}), \mathbf{x} - \mathbf{y} \rangle . \tag{11}$$

**Definition 2** ($L$-smooth). *The function $f : \mathbb{R}^d \to \mathbb{R}$, is L-smooth, if there exists a constant $L > 0$ such that for all $\mathbf{x}, \mathbf{y} \in \mathbb{R}^d$*

$$\|\nabla f(\mathbf{x}) - \nabla f(\mathbf{y})\| \leq L\|\mathbf{x} - \mathbf{y}\|, \tag{12}$$

*or equivalently (Nesterov, 2003, Lemma 1.2.3)*

$$f(\mathbf{x}) \leq f(\mathbf{y}) + \langle \nabla f(\mathbf{y}), \mathbf{x} - \mathbf{y} \rangle + \frac{L}{2} \|\mathbf{x} - \mathbf{y}\|^2 . \tag{13}$$

**Lemma 7** (Li & Orabona (2019), Lemma 4). *For a L-smooth function $f : \mathbb{R}^d \to \mathbb{R}$, having an optima at $\mathbf{x}^\star \in \mathbb{R}^d$, we have for any $\mathbf{x} \in \mathbb{R}^d$*

$$\|\nabla f(\mathbf{x}) - \nabla f(\mathbf{x}^\star)\|^2 \leq 2L \left( f(\mathbf{x}) - f(\mathbf{x}^\star) \right) . \tag{14}$$

**Lemma 8** (Orvieto et al. (2022), Lemma B.2). *For a L-smooth and $\mu-$strongly convex function $f : \mathbb{R}^d \to \mathbb{R}$, the following bound holds*

$$\frac{1}{2cL} \leq \frac{f(\mathbf{x}) - f^\star}{c \|\nabla f(\mathbf{x})\|^2} \leq \frac{f(\mathbf{x}) - \ell^\star}{c \|\nabla f(\mathbf{x})\|^2} \leq \frac{1}{2c\mu} , \tag{15}$$

*where $f^\star = \inf_{\mathbf{x}} f(\mathbf{x})$, and $\ell^\star$ is a lower bound $\ell^\star \leq f^\star$.*

### A.2  GENERAL INEQUALITIES

**Lemma 9.** *For arbitrary set of $n$ vectors $\{\mathbf{a}_i\}_{i=1}^n$, $\mathbf{a}_i \in \mathbb{R}^d$*

$$\frac{1}{n} \sum_{i=1}^n \left\| \mathbf{a}_i - \left( \frac{1}{n} \sum_{i=1}^n \mathbf{a}_i \right) \right\|^2 \leq \frac{1}{n} \sum_{i=1}^n \|\mathbf{a}_i\|^2 . \tag{16}$$

**Lemma 10.** *For arbitrary set of $n$ vectors $\{\mathbf{a}_i\}_{i=1}^n$, $\mathbf{a}_i \in \mathbb{R}^d$*

$$\left\| \sum_{i=1}^n \mathbf{a}_i \right\|^2 \leq n \sum_{i=1}^n \|\mathbf{a}_i\|^2 . \tag{17}$$

**Lemma 11.** *For given two vectors $\mathbf{a}, \mathbf{b} \in \mathbb{R}^d$*

$$2 \langle \mathbf{a}, \mathbf{b} \rangle \leq \gamma \|\mathbf{a}\|^2 + \gamma^{-1} \|\mathbf{b}\|^2 , \qquad\qquad \forall \gamma > 0 . \tag{18}$$

**Lemma 12.** *For given two vectors $\mathbf{a}, \mathbf{b} \in \mathbb{R}^d$*

$$\|\mathbf{a} + \mathbf{b}\|^2 \leq (1 + \lambda) \|\mathbf{a}\|^2 + (1 + \lambda^{-1}) \|\mathbf{b}\|^2 , \qquad\qquad \forall \lambda > 0 . \tag{19}$$

**Lemma 13.** *For any random variable $X$*

$$\mathbb{E} \|X - \mathbb{E}[X]\|^2 \leq \mathbb{E} \|X\|^2 . \tag{20}$$

## B  CONVERGENCE ANALYSIS OF FEDSPS

### B.1  FEDSPS INEQUALITIES

**Lemma 14** (Upper and lower bounds on FedSPS stepsizes). *Under the assumption that each $F_i$ is L-smooth (Assumption 1), the stepsizes of FedSPS follow for all rounds $t \in [T-1]$*

$$\alpha \leq \gamma_t \leq \gamma_b, \tag{21}$$

*where $\alpha = \min\left\{\frac{1}{2cL}, \gamma_b\right\}$.*

*Proof.* This lemma is easily obtained by plugging in the definition of FedSPS stepsizes into (15). $\square$

**Lemma 15.** *FedSPS stepsizes $\gamma_t^i$ follow the following fundamental inequality*

$$\left\|\gamma_t^i \mathbf{g}_t^i\right\|^2 \leq \frac{\gamma_t^i}{c}[F_i(\mathbf{x}_t^i, \xi_t^i) - \ell_i^\star] \tag{22}$$

*Proof.* We observe that it holds from the definition of FedSPS stepsizes

$$\left\|\gamma_t^i \mathbf{g}_t^i\right\|^2 = \gamma_t^i \cdot \min\left\{\frac{F(\mathbf{x}_t^i, \xi_t^i) - \ell_i^\star}{c\left\|\nabla F(\mathbf{x}_t^i, \xi_t^i)\right\|^2}, \gamma_b\right\}\left\|\nabla F_i(\mathbf{x}_t^i, \xi_t^i)\right\|^2$$

$$\leq \frac{\gamma_t^i}{c}[F_i(\mathbf{x}_t^i, \xi_t^i) - \ell_i^\star]. \qquad\square$$

### B.2  PROOF FOR CONVEX CASE OF FEDSPS

**Proof outline and notation.** Before we start with the proof, let us recall the notation: below we will frequently use $\mathbf{g}_t^i$ to denote the random component sampled by worker $i$ at iteration $t$, so $\mathbf{g}_t^i = \nabla F_i(\mathbf{x}_t^i, \xi_t^i)$ and the stepsize $\gamma_t^i = \min\left\{\frac{F_i(\mathbf{x}_t^i, \xi_t^i) - \ell_i^\star}{c\left\|\nabla F_i(\mathbf{x}_t^i, \xi_t^i)\right\|^2}, \gamma_b\right\}$. We define the (possibly virtual) average iterate $\bar{\mathbf{x}}_t := \frac{1}{n}\sum_{i=1}^n \mathbf{x}_t^i$. We follow the proof template for FedAvg (Local SGD) Koloskova et al. (2020); Stich (2019), deriving a difference lemma to bound $R_t := \frac{1}{n}\sum_{i=1}^n \left\|\bar{\mathbf{x}}_t - \mathbf{x}_t^i\right\|^2$, and plugging it into the decrease $\|\bar{\mathbf{x}}_{t+1} - \mathbf{x}^\star\|^2$. Our poof introduces differences at various points like in the difference lemma (25), and the decrease (29), to incorporate local adaptivity using properties of FedSPS stepsizes (Lemma 14 and Lemma 15) and the finite optimal objective difference assumption ($\sigma_f^2 < \infty$).

#### B.2.1  DIFFERENCE LEMMAS

**Lemma 16** (Bound on difference $R_t$ for convex case with any stepsizes). *For $c \geq 2\tau^2$, the variance of the iterates between the clients $R_t := \frac{1}{n}\sum_{i=1}^n \left\|\bar{\mathbf{x}}_t - \mathbf{x}_t^i\right\|^2$, is bounded as*

$$R_t \leq \frac{1}{2n\tau}\sum_{i=1}^n \sum_{j=(t-1)-k(t)}^{t-1} \gamma_j^i[F_i(\mathbf{x}_j^i, \xi_j^i) - \ell_i^\star], \tag{23}$$

*where $t - 1 - k(t)$ denotes the index of the last communication round ($k \leq \tau - 1$).*

*Proof.* We use the property that $\bar{\mathbf{x}}_t = \mathbf{x}_t^i$ for every $t$ that is a multiple of $\tau$. Therefore, there exists a $k(t) \leq \tau - 1$ such that $R_{(t-1)-k(t)} = 0$. So we have,

$$R_t = \frac{1}{n}\sum_{i=1}^n \left\|\sum_{j=(t-1)-k(t)}^{t-1} \gamma_j^i \mathbf{g}_j^i - \mathbf{v}\right\|^2 \overset{(16)}{\leq} \frac{1}{n}\sum_{i=1}^n \left\|\sum_{j=(t-1)-k(t)}^{t-1} \gamma_j^i \mathbf{g}_j^i\right\|^2,$$

where $\mathbf{v} := \frac{1}{n} \sum_{i=1}^{n} \sum_{j=(t-1)-k(t)}^{t-1} \gamma_j^i \mathbf{g}_j^i$. The inequality follows by the fact the $\mathbf{v}$ is the mean of the terms in the sum. With the inequality $\sum_{i=1}^{\tau} \|\mathbf{a}_i\|^2 \leq \tau \sum_{i=1}^{\tau} \|\mathbf{a}_i\|^2$ for vectors $\mathbf{a}_i \in \mathbb{R}^d$, and the property of the Polyak Stepsize we deduce:

$$R_t \overset{(17)}{\leq} \frac{\tau}{n} \sum_{i=1}^{n} \sum_{j=(t-1)-k(t)}^{t-1} \left\| \gamma_j^i \mathbf{g}_j^i \right\|^2 \tag{24}$$

$$\overset{(22)}{\leq} \frac{\tau}{nc} \sum_{i=1}^{n} \sum_{j=(t-1)-k(t)}^{t-1} \gamma_j^i [F_i(\mathbf{x}_j^i, \xi_j^i) - \ell_i^\star] \tag{25}$$

$$\leq \frac{1}{2n\tau} \sum_{i=1}^{n} \sum_{j=(t-1)-k(t)}^{t-1} \gamma_j^i [F_i(\mathbf{x}_j^i, \xi_j^i) - \ell_i^\star] \,,$$

where we used the assumption $c \geq 2\tau^2$ to obtain the last inequality. $\qquad\square$

**Lemma 17** (Bound on difference $R_t$ for convex case with small stepsizes). *For $\gamma_b \leq \frac{1}{20L\tau}$, the variance of the iterates between the clients $R_t := \frac{1}{n} \sum_{i=1}^{n} \left\| \bar{\mathbf{x}}_t - \mathbf{x}_t^i \right\|^2$, is bounded as*

$$\mathbb{E}\, R_t \leq \frac{1}{3L\tau} \sum_{j=(t-1)-k(t)}^{t-1} \mathbb{E}[f(\bar{\mathbf{x}}_j) - f^\star] + 64L\gamma_b^2 \tau \sum_{j=(t-1)-k(t)}^{t-1} \sigma_f^2 \,, \tag{26}$$

*where $t - 1 - k(t)$ denotes the index of the last communication round ($k \leq \tau - 1$).*

*Proof.* Again we derive a bound on the variance on the iterated between the clients. We use the property that $\bar{\mathbf{x}}_t = \mathbf{x}_t^i$ for every $t$ that is a multiple of $\tau$. Define $k(t) \leq \tau - 1$ as above. We will now prove that it holds

$$R_t \leq \frac{32\gamma_b^2 \tau}{n} \sum_{i=1}^{n} \sum_{j=(t-1)-k(t)}^{t-1} \left\| \nabla F_i(\bar{\mathbf{x}}_j, \xi_j^i) \right\|^2 \tag{27}$$

$$\overset{(12)}{\leq} \frac{64\gamma_b^2 L\tau}{n} \sum_{i=1}^{n} \sum_{j=(t-1)-k(t)}^{t-1} [F_i(\bar{\mathbf{x}}_j, \xi_j^i) - \ell_i^\star] \,,$$

and consequently (after taking expectation) and using $\gamma_b \leq \frac{1}{20L\tau}$:

$$\mathbb{E}\, R_t \leq \frac{1}{3L\tau} \sum_{j=(t-1)-k(t)}^{t-1} \mathbb{E}[f(\bar{\mathbf{x}}_j) - f^\star] + 64L\gamma_b^2 \tau \sum_{j=(t-1)-k(t)}^{t-1} \sigma_f^2 \,.$$

Note that if $t$ is a multiple of $\tau$, then $R_t = 0$ and there is nothing to prove. Otherwise note that

$$R_{t+1} = \frac{1}{n} \sum_{i=1}^{n} \left\| \bar{\mathbf{x}}_t - \mathbf{x}_t^i + \gamma_t^i \nabla F(\mathbf{x}_t^i, \xi_t^i) - \mathbf{v} \right\|^2 \,,$$

where $\mathbf{v} := \frac{1}{n}\sum_{i=1}^{n}\gamma_t^i\nabla F(\mathbf{x}_t^i,\xi_t^i)$ denotes the average of the client updates. With the inequality $\|\mathbf{a}+\mathbf{b}\|^2 \le (1+\tau^{-1})\|\mathbf{a}\|^2 + 2\tau\|\mathbf{b}\|^2$ for $\tau \ge 1$, $\mathbf{a},\mathbf{b}\in\mathbb{R}^d$, we continue:

$$
\begin{aligned}
R_{t+1} &\overset{(19)}{\le} \left(1+\frac{1}{\tau}\right)R_t + \frac{2\tau}{n}\sum_{i=1}^{n}\left\|\gamma_t^i\nabla F_i(\mathbf{x}_t^i,\xi_t^i)-\mathbf{v}\right\|^2 \\
&\overset{(16)}{\le} \left(1+\frac{1}{\tau}\right)R_t + \frac{2\tau}{n}\sum_{i=1}^{n}\left\|\gamma_t^i\nabla F_i(\mathbf{x}_t^i,\xi_t^i)\right\|^2 \\
&\le \left(1+\frac{1}{\tau}\right)R_t + \frac{2\tau\gamma_b^2}{n}\sum_{i=1}^{n}\left\|\nabla F_i(\mathbf{x}_t^i,\xi_t^i)\right\|^2 \\
&\overset{(19)}{\le} \left(1+\frac{1}{\tau}\right)R_t + \frac{4\tau\gamma_b^2}{n}\sum_{i=1}^{n}\left(\left\|\nabla F_i(\mathbf{x}_t^i,\xi_t^i)-\nabla F_i(\bar{\mathbf{x}}_t,\xi_t^i)\right\|^2 + \left\|\nabla F_i(\bar{\mathbf{x}}_t,\xi_t^i)\right\|^2\right) \\
&\overset{(12)}{\le} \left(1+\frac{1}{\tau}\right)R_t + \frac{4\tau\gamma_b^2}{n}\sum_{i=1}^{n}\left(L^2\left\|\bar{\mathbf{x}}_t-\mathbf{x}_t^i\right\|^2 + \left\|\nabla F_i(\bar{\mathbf{x}}_t,\xi_t^i)\right\|^2\right) \\
&\le \left(1+\frac{2}{\tau}\right)R_t + \frac{4\tau\gamma_b^2}{n}\sum_{i=1}^{n}\left\|\nabla F_i(\bar{\mathbf{x}}_t,\xi_t^i)\right\|^2\,,
\end{aligned}
$$

where we used $\gamma_b \le \frac{1}{10L\tau}$. Equation (27) now follows by unrolling, and noting that $\left(1+\frac{2}{\tau}\right)^j \le 8$ for all $0 \le j \le \tau$. $\qquad\square$

### B.2.2 PROOF OF THEOREM 3 (A) (GENERAL CASE VALID FOR ALL STEPSIZES)

**Distance to optimality.** We now proceed by using the definition of the virtual average: $\bar{\mathbf{x}}_{t+1} = \frac{1}{n}\sum_{i=1}^{n}\left(\mathbf{x}_t^i-\gamma_t^i\mathbf{g}_t^i\right)$.

$$
\begin{aligned}
\|\bar{\mathbf{x}}_{t+1}-\mathbf{x}^\star\|^2 &\le \|\bar{\mathbf{x}}_t-\mathbf{x}^\star\|^2 - \frac{2}{n}\sum_{i=1}^{n}\left\langle\bar{\mathbf{x}}_t-\mathbf{x}^\star,\gamma_t^i\mathbf{g}_t^i\right\rangle + \frac{1}{n}\sum_{i=1}^{n}\left\|\gamma_t^i\mathbf{g}_t^i\right\|^2 \\
&= \|\bar{\mathbf{x}}_t-\mathbf{x}^\star\|^2 - \frac{2}{n}\sum_{i=1}^{n}\left\langle\mathbf{x}_t^i-\mathbf{x}^\star,\gamma_t^i\mathbf{g}_t^i\right\rangle + \frac{1}{n}\sum_{i=1}^{n}\left\|\gamma_t^i\mathbf{g}_t^i\right\|^2 - \frac{2}{n}\sum_{i=1}^{n}\left\langle\bar{\mathbf{x}}_t-\mathbf{x}_t^i,\gamma_t^i\mathbf{g}_t^i\right\rangle \\
&\overset{(18)}{\le} \|\bar{\mathbf{x}}_t-\mathbf{x}^\star\|^2 - \frac{2}{n}\sum_{i=1}^{n}\left\langle\mathbf{x}_t^i-\mathbf{x}^\star,\gamma_t^i\mathbf{g}_t^i\right\rangle + \frac{1}{n}\sum_{i=1}^{n}\left\|\gamma_t^i\mathbf{g}_t^i\right\|^2 + \frac{1}{n}\sum_{i=1}^{n}\left(\left\|\bar{\mathbf{x}}_t-\mathbf{x}_t^i\right\|^2 + \left\|\gamma_t^i\mathbf{g}_t^i\right\|^2\right) \\
&= \|\bar{\mathbf{x}}_t-\mathbf{x}^\star\|^2 - \frac{2}{n}\sum_{i=1}^{n}\left\langle\mathbf{x}_t^i-\mathbf{x}^\star,\gamma_t^i\mathbf{g}_t^i\right\rangle + \frac{2}{n}\sum_{i=1}^{n}\left\|\gamma_t^i\mathbf{g}_t^i\right\|^2 + R_t\,. \qquad (28)
\end{aligned}
$$

**Upper bound (valid for arbitrary stepsizes).** We now proceed in a similar fashion as outlined in the proof of (Loizou et al., 2021, Theorem 3.4). Using the property (22) of the FeSPS stepsize we

get:

$$\|\bar{\mathbf{x}}_{t+1} - \mathbf{x}^\star\|^2 \overset{(22)}{\leq} \|\bar{\mathbf{x}}_t - \mathbf{x}^\star\|^2 - \frac{2}{n}\sum_{i=1}^n \langle \mathbf{x}_t^i - \mathbf{x}^\star, \gamma_t^i \mathbf{g}_t^i \rangle + \frac{2}{nc}\sum_{i=1}^n \gamma_t^i [F_i(\mathbf{x}_t^i, \xi_t^i) - \ell_i^\star] + R_t \quad (29)$$

$$\overset{(11)}{\leq} \|\bar{\mathbf{x}}_t - \mathbf{x}^\star\|^2 - \frac{2}{n}\sum_{i=1}^n \gamma_t^i [F_i(\mathbf{x}_t^i, \xi_t^i) - F_i(\mathbf{x}^\star, \xi_t^i)] + \frac{2}{nc}\sum_{i=1}^n \gamma_t^i [F_i(\mathbf{x}_t^i, \xi_t^i) - \ell_i^\star] + R_t$$

$$= \|\bar{\mathbf{x}}_t - \mathbf{x}^\star\|^2 - \frac{2}{n}\sum_{i=1}^n \gamma_t^i [F_i(\mathbf{x}_t^i, \xi_t^i) - \ell_i^\star + \ell_i^\star - F_i(\mathbf{x}^\star, \xi_t^i)]$$

$$+ \frac{2}{nc}\sum_{i=1}^n \gamma_t^i [F_i(\mathbf{x}_t^i, \xi_t^i) - \ell_i^\star] + R_t$$

$$= \|\bar{\mathbf{x}}_t - \mathbf{x}^\star\|^2 - \left(2 - \frac{2}{c}\right)\frac{1}{n}\sum_{i=1}^n \gamma_t^i [F_i(\mathbf{x}_t^i, \xi_t^i) - \ell_i^\star] + \frac{2}{n}\sum_{i=1}^n \gamma_t^i [F_i(\mathbf{x}^\star, \xi_t^i) - \ell_i^\star] + R_t \,,$$

$$(30)$$

We use the assumption $c > 2$ and simplify further:[2]

$$\|\bar{\mathbf{x}}_{t+1} - \mathbf{x}^\star\|^2 \leq \|\bar{\mathbf{x}}_t - \mathbf{x}^\star\|^2 - \frac{1}{n}\sum_{i=1}^n \gamma_t^i [F_i(\mathbf{x}_t^i, \xi_t^i) - \ell_i^\star] + 2\gamma_b \sigma_t^2 + R_t \,,$$

where we defined $\sigma_t^2 := \frac{1}{n}\sum_{i=1}^n [F_i(\mathbf{x}^\star, \xi_t^i) - \ell_i^\star]$. After rearranging:

$$\frac{1}{n}\sum_{i=1}^n \gamma_t^i [F_i(\mathbf{x}_t^i, \xi_t^i) - \ell_i^\star] \leq \|\bar{\mathbf{x}}_t - \mathbf{x}^\star\|^2 - \|\bar{\mathbf{x}}_{t+1} - \mathbf{x}^\star\|^2 + 2\gamma_b \sigma_t^2 + R_t \,.$$

We now plug in the bound on $R_t$ calculated above in equation (23):

$$\frac{1}{n}\sum_{i=1}^n \gamma_t^i [F_i(\mathbf{x}_t^i, \xi_t^i) - \ell_i^\star] \leq \|\bar{\mathbf{x}}_t - \mathbf{x}^\star\|^2 - \|\bar{\mathbf{x}}_{t+1} - \mathbf{x}^\star\|^2 + 2\gamma_b \sigma_t^2 + \frac{1}{2n\tau}\sum_{i=1}^n \sum_{j=(t-1)-k(t)}^{t-1} \gamma_j^i [F_i(\mathbf{x}_j^i, \xi_j^i) - \ell_i^\star]$$

$$\leq \|\bar{\mathbf{x}}_t - \mathbf{x}^\star\|^2 - \|\bar{\mathbf{x}}_{t+1} - \mathbf{x}^\star\|^2 + 2\gamma_b \sigma_t^2 + \frac{1}{2n\tau}\sum_{i=1}^n \sum_{j=(t-1)-(\tau-1)}^{t-1} \gamma_j^i [F_i(\mathbf{x}_j^i, \xi_j^i) - \ell_i^\star] \,.$$

We now sum this equation up from $t = 0$ to $T - 1$, and divide by $T$:

$$\frac{1}{T}\sum_{t=0}^{T-1}\frac{1}{n}\sum_{i=1}^n \gamma_t^i [F_i(\mathbf{x}_t^i, \xi_t^i) - \ell_i^\star] \leq \frac{1}{T}\sum_{t=0}^{T-1}\left(\|\bar{\mathbf{x}}_t - \mathbf{x}^\star\|^2 - \|\bar{\mathbf{x}}_{t+1} - \mathbf{x}^\star\|^2\right)$$

$$+ 2\gamma_b \frac{1}{T}\sum_{t=0}^{T-1}\sigma_t^2 + \frac{1}{T}\sum_{t=0}^{T-1}\frac{1}{2n}\sum_{i=1}^n \gamma_t^i [F_i(\mathbf{x}_t^i, \xi_j^t) - \ell_i^\star] \,,$$

by noting that each component in the last term appears at most $\tau$ times in the sum. We can now move the last term to the left:

$$\frac{1}{T}\sum_{t=0}^{T-1}\frac{1}{2n}\sum_{i=1}^n \gamma_t^i [F_i(\mathbf{x}_t^i, \xi_t^i) - \ell_i^\star] \leq \frac{1}{T}\|\bar{\mathbf{x}}_0 - \mathbf{x}^\star\|^2 + 2\gamma_b \frac{1}{T}\sum_{t=0}^{T-1}\sigma_t^2 \,.$$

It remains to note $\gamma_t^i \geq \alpha := \min\left\{\frac{1}{2cL}, \gamma_b\right\}$, therefore:

$$\frac{1}{n}\sum_{i=1}^n \gamma_t^i [F_i(\mathbf{x}_t^i, \xi_t^i) - \ell_i^\star] \geq \frac{1}{n}\sum_{i=1}^n \alpha [F_i(\mathbf{x}_t^i, \xi_t^i) - \ell_i^\star] \,.$$

---

[2] The attentive reader will observe that any $c > \frac{1}{2}$ would be sufficient to show convergence of the function suboptimality (note that Loizou et al. (2021) used $c \geq \frac{1}{2}$, but only showed convergence of the iterate distance to optimality).

To summarize:

$$\frac{1}{T}\sum_{t=0}^{T-1}\frac{1}{2n}\sum_{i=1}^{n}[F_i(\mathbf{x}_t^i,\xi_t^i)-\ell_i^\star] \le \frac{1}{T\alpha}\left\|\bar{\mathbf{x}}_0-\mathbf{x}^\star\right\|^2 + \frac{2\gamma_b}{\alpha}\frac{1}{T}\sum_{t=0}^{T-1}\sigma_t^2 . \tag{31}$$

We now take full expectation to get the claimed result.

### B.2.3 Proof of Theorem 6 (a) (special case with small step sizes)

We now give an additional bound that tightens the result when the parameter $\gamma_b$ is chosen to be a small value, smaller than $\frac{1}{2cL}$. As a consequence of this assumption (see Equation (15)), it holds $\gamma_t^i \equiv \gamma_b$ for all $t$ and $i \in [n]$. The proof now follows a similar template as above, but we can now directly utilize the imposed upper bound on $\gamma_b$ (v.s. the bound on $c$ used previously).

**Upper bound (valid for small stepsizes).** Using the definition of the virtual average: $\bar{\mathbf{x}}_{t+1} = \frac{1}{n}\sum_{i=1}^{n}\left(\mathbf{x}_t^i-\gamma_t^i\mathbf{g}_t^i\right)$.

$$\left\|\bar{\mathbf{x}}_{t+1}-\mathbf{x}^\star\right\|^2 \le \left\|\bar{\mathbf{x}}_t-\mathbf{x}^\star\right\|^2 - \frac{2}{n}\sum_{i=1}^{n}\left\langle\bar{\mathbf{x}}_t-\mathbf{x}^\star,\gamma_t^i\mathbf{g}_t^i\right\rangle + \frac{1}{n}\sum_{i=1}^{n}\left\|\gamma_t^i\mathbf{g}_t^i\right\|^2 \tag{32}$$

We now observe

$$-\left\langle\bar{\mathbf{x}}_t-\mathbf{x}^\star,\gamma_t^i\mathbf{g}_t^i\right\rangle = -\gamma_t^i\left\langle\bar{\mathbf{x}}_t-\mathbf{x}_t^i,\mathbf{g}_t^i\right\rangle - \gamma_t^i\left\langle\mathbf{x}_t^i-\mathbf{x}^\star,\mathbf{g}_t^i\right\rangle$$

$$\overset{(13)}{\le} -\gamma_t^i\left[F_i(\bar{\mathbf{x}}_t,\xi_t^i)-F_i(\mathbf{x}_t^i,\xi_t^i)-\frac{L}{2}\left\|\bar{\mathbf{x}}_t-\mathbf{x}_t^i\right\|^2\right] - \gamma_t^i\left\langle\mathbf{x}_t^i-\mathbf{x}^\star,\mathbf{g}_t^i\right\rangle \tag{33}$$

$$\overset{(11)}{\le} -\gamma_t^i\left[F_i(\bar{\mathbf{x}}_t,\xi_t^i)-F_i(\mathbf{x}_t^i,\xi_t^i)-\frac{L}{2}\left\|\bar{\mathbf{x}}_t-\mathbf{x}_t^i\right\|^2+F_i(\mathbf{x}_t^i,\xi_t^i)-F_i(\mathbf{x}^\star,\xi_t^i)\right]$$

$$= -\gamma_t^i\left[F_i(\bar{\mathbf{x}}_t,\xi_t^i)-F_i(\mathbf{x}^\star,\xi_t^i)\right] + \frac{\gamma_t^i L}{2}\left\|\bar{\mathbf{x}}_t-\mathbf{x}_t^i\right\|^2 . \tag{34}$$

Therefore

$$\left\|\bar{\mathbf{x}}_{t+1}-\mathbf{x}^\star\right\|^2 \overset{(32),(34)}{\le} \left\|\bar{\mathbf{x}}_t-\mathbf{x}^\star\right\|^2 - \frac{2}{n}\sum_{i=1}^{n}\gamma_t^i[F_i(\bar{\mathbf{x}}_t,\xi_t^i)-F_i(\mathbf{x}^\star,\xi_t^i)] + \frac{L}{n}\sum_{i=1}^{n}\gamma_t^i\left\|\bar{\mathbf{x}}_t-\mathbf{x}_t^i\right\|^2 + \frac{1}{n}\sum_{i=1}^{n}\left\|\gamma_t^i\mathbf{g}_t^i\right\|^2 . \tag{35}$$

We now use the observation that $\gamma_t^i = \gamma_b$ (by the assumption that $\gamma_b$ is small—we could have made this simplification also earlier), and continue:

$$\left\|\bar{\mathbf{x}}_{t+1}-\mathbf{x}^\star\right\|^2 \le \left\|\bar{\mathbf{x}}_t-\mathbf{x}^\star\right\|^2 - \frac{2\gamma_b}{n}\sum_{i=1}^{n}[F_i(\bar{\mathbf{x}}_t,\xi_t^i)-F_i(\mathbf{x}^\star,\xi_t^i)] + \gamma_b L R_t + \frac{\gamma_b^2}{n}\sum_{i=1}^{n}\left\|\mathbf{g}_t^i\right\|^2 .$$

We now use

$$\left\|\mathbf{g}_t^i\right\|^2 \overset{(19)}{\le} 2\left\|\nabla F_i(\bar{\mathbf{x}}_t,\xi_t^i)-\nabla F_i(\mathbf{x}_t^i,\xi_t^i)\right\|^2 + 2\left\|\nabla F_i(\bar{\mathbf{x}}_t,\xi_t^i)\right\|^2$$

$$\overset{(12)}{\le} 2L^2\left\|\bar{\mathbf{x}}_t-\mathbf{x}_t^i\right\|^2 + 2\left\|\nabla F_i(\bar{\mathbf{x}}_t,\xi_t^i)\right\|^2$$

and the assumption $\gamma_b \le \frac{1}{10L}$, to obtain

$$\left\|\bar{\mathbf{x}}_{t+1}-\mathbf{x}^\star\right\|^2 \le \left\|\bar{\mathbf{x}}_t-\mathbf{x}^\star\right\|^2 - \frac{2\gamma_b}{n}\sum_{i=1}^{n}[F_i(\bar{\mathbf{x}}_t,\xi_t^i)-F_i(\mathbf{x}^\star,\xi_t^i)] + 2\gamma_b L R_t + \frac{2\gamma_b^2}{n}\sum_{i=1}^{n}\left\|\nabla F_i(\bar{\mathbf{x}}_t,\xi_t^i)\right\|^2$$

$$\le \left\|\bar{\mathbf{x}}_t-\mathbf{x}^\star\right\|^2 - \frac{2\gamma_b}{n}\sum_{i=1}^{n}[F_i(\bar{\mathbf{x}}_t,\xi_t^i)-F_i(\mathbf{x}^\star,\xi_t^i)] + 2\gamma_b L R_t + \frac{4L\gamma_b^2}{n}\sum_{i=1}^{n}[F_i(\bar{\mathbf{x}}_t,\xi_t^i)-\ell_i^\star] .$$

To simplify, let's take expectation over the randomness in iteration $t$:

$$\mathbb{E}\left\|\bar{\mathbf{x}}_{t+1}-\mathbf{x}^\star\right\|^2 \le \left\|\bar{\mathbf{x}}_t-\mathbf{x}^\star\right\|^2 - 2\gamma_b[f(\bar{\mathbf{x}}_t)-f^\star] + 2\gamma_b L\,\mathbb{E}[R_t] + 4L\gamma_b^2[f(\bar{\mathbf{x}}_t)-f^\star+\sigma_f^2] .$$

Because $f(\bar{\mathbf{x}}_t) - f^\star \geq 0$ and $\gamma_b \leq \frac{1}{10L}$, we can further simplify:

$$\mathbb{E}\left\|\bar{\mathbf{x}}_{t+1} - \mathbf{x}^\star\right\|^2 \leq \left\|\bar{\mathbf{x}}_t - \mathbf{x}^\star\right\|^2 - \gamma_b[f(\bar{\mathbf{x}}_t) - f^\star] + 2\gamma_b L\,\mathbb{E}[R_t] + 4L\gamma_b^2\sigma_f^2. \qquad (36)$$

After re-arranging and taking expectation:

$$\gamma_b\,\mathbb{E}[f(\bar{\mathbf{x}}_t) - f^\star] \leq \mathbb{E}\left\|\bar{\mathbf{x}}_t - \mathbf{x}^\star\right\|^2 - \mathbb{E}\left\|\bar{\mathbf{x}}_{t+1} - \mathbf{x}^\star\right\|^2 + 2\gamma_b L\,\mathbb{E}[R_t] + 4L\gamma_b^2\sigma_f^2.$$

We now plug in the bound on $R_t$ from Equation (26):

$$\gamma_b\,\mathbb{E}[f(\bar{\mathbf{x}}_t) - f^\star] \leq \mathbb{E}\left\|\bar{\mathbf{x}}_t - \mathbf{x}^\star\right\|^2 - \mathbb{E}\left\|\bar{\mathbf{x}}_{t+1} - \mathbf{x}^\star\right\|^2 + \frac{2\gamma_b}{3\tau}\sum_{j=(t-1)-k(t)}^{t-1}[f(\bar{\mathbf{x}}_j) - f^\star] + 4(\gamma_b^2 + 32\tau^2\gamma_b^3)L\sigma_f^2,$$

and after summing over $t = 0$ to $T - 1$, and dividing by $T$:

$$\frac{1}{T}\sum_{t=0}^{T-1}\gamma_b\,\mathbb{E}[f(\bar{\mathbf{x}}_t) - f^\star] \leq \frac{1}{T}\sum_{t=0}^{T-1}\left(\mathbb{E}\left\|\bar{\mathbf{x}}_t - \mathbf{x}^\star\right\|^2 - \mathbb{E}\left\|\bar{\mathbf{x}}_{t+1} - \mathbf{x}^\star\right\|^2\right) + \frac{2\gamma_b}{3T}\sum_{t=0}^{T-1}[f(\bar{\mathbf{x}}_t) - f^\star] + 4(\gamma_b^2 + 32\tau^2\gamma_b^3)L\sigma_f^2,$$

and consequently:

$$\frac{1}{T}\sum_{t=0}^{T-1}\mathbb{E}[f(\bar{\mathbf{x}}_t) - f^\star] \leq \frac{3}{T\gamma_b}\mathbb{E}\left\|\bar{\mathbf{x}}_0 - \mathbf{x}^\star\right\|^2 + 12(\gamma_b + 32\tau^2\gamma_b^2)L\sigma_f^2. \qquad (37)$$

### B.3 Proof for strongly convex case of FedSPS

Here we outline the proof in the strongly convex case. As the proof is closely following the arguments from the convex case (just with the difference of applying $\mu$-strong convexity whenever convexity was used), we just highlight the main differences for the readers.

#### B.3.1 Proof of Theorem 3 (b) (general case valid for all stepsizes)

In the respective proof in the convex case, convexity was used in Equation (29). If we use strong convexity instead, we obtain

$$\left\|\bar{\mathbf{x}}_{t+1} - \mathbf{x}^\star\right\|^2 \overset{(4)}{\leq} \left\|\bar{\mathbf{x}}_t - \mathbf{x}^\star\right\|^2 - \frac{2}{n}\sum_{i=1}^n \gamma_t^i[F_i(\mathbf{x}_t^i, \xi_t^i) - F_i(\mathbf{x}^\star, \xi_t^i) + \frac{\mu}{2}\left\|\mathbf{x}_t^i - \mathbf{x}^\star\right\|^2]$$

$$+ \frac{2}{nc}\sum_{i=1}^n \gamma_t^i[F_i(\mathbf{x}_t^i, \xi_t^i) - \ell_i^\star] + R_t$$

$$\leq \left\|\bar{\mathbf{x}}_t - \mathbf{x}^\star\right\|^2 - \frac{2}{n}\sum_{i=1}^n \gamma_t^i[F_i(\mathbf{x}_t^i, \xi_t^i) - F_i(\mathbf{x}^\star, \xi_t^i) + \frac{\mu}{2}\left\|\bar{\mathbf{x}}_t - \mathbf{x}^\star\right\|^2] + \frac{2}{n}\sum_{i=1}^n \gamma_t^i\mu\left\|\bar{\mathbf{x}}_t - \mathbf{x}_t^i\right\|^2$$

$$+ \frac{2}{nc}\sum_{i=1}^n \gamma_t^i[F_i(\mathbf{x}_t^i, \xi_t^i) - \ell_i^\star] + R_t$$

$$\leq \left\|\bar{\mathbf{x}}_t - \mathbf{x}^\star\right\|^2 - \frac{2}{n}\sum_{i=1}^n \gamma_t^i[F_i(\mathbf{x}_t^i, \xi_t^i) - F_i(\mathbf{x}^\star, \xi_t^i) + \frac{\mu}{2}\left\|\bar{\mathbf{x}}_t - \mathbf{x}^\star\right\|^2]$$

$$+ \frac{2}{nc}\sum_{i=1}^n \gamma_t^i[F_i(\mathbf{x}_t^i, \xi_t^i) - \ell_i^\star] + 2R_t$$

where the second inequality used $\left\|\bar{\mathbf{x}}_t - \mathbf{x}^\star\right\|^2 \leq 2\left\|\bar{\mathbf{x}}_t - \mathbf{x}_t^i\right\|^2 + 2\left\|\mathbf{x}_t^i - \mathbf{x}^\star\right\|^2$ and the last inequality used $\gamma_t^i\mu \leq \frac{1}{2c} \leq \frac{1}{2}$. By applying the lower bound $\alpha$ on the stepsize, we see that convergence is governed by linearly decreasing term:

$$\left\|\bar{\mathbf{x}}_{t+1} - \mathbf{x}^\star\right\|^2 \leq (1 - \mu\alpha)\left\|\bar{\mathbf{x}}_t - \mathbf{x}^\star\right\|^2 - \frac{2}{n}\sum_{i=1}^n \gamma_t^i[F_i(\mathbf{x}_t^i, \xi_t^i) - F_i(\mathbf{x}^\star, \xi_t^i)] + \frac{2}{nc}\sum_{i=1}^n \gamma_t^i[F_i(\mathbf{x}_t^i, \xi_t^i) - \ell_i^\star] + 2R_t,$$

while the remaining terms are the same (up to a small change in the constant in front of $R_t$) as in the convex case. The increased constant can be handled by imposing a slightly stronger condition on $c$ (here we used $c \geq 4\tau^2$, compared to $c \geq 2\tau^2$ in the convex case). The rest of the proof follows by the same steps, with one additional departure: when summing up the inequalities over the iterations $t = 0$ to $T - 1$ we need to use an appropriate weighing to benefit from the linear decrease. This technique is standard in the analysis (illustrated in Stich (2019) and carried out in the setting a residual $R_t$ in (Stich & Karimireddy, 2020, Proof of Theorem 16) and with a residual $R_t$ in a distributed setting in (Koloskova et al., 2020, Proof of Theorem 2, in particular Lemma 15).

### B.3.2 Proof of Theorem 6 (b) (special case with small step sizes)

Convexity was used in Equation (33). If we use $\mu$-strong convexity instead, we obtain:

$$-\left\langle \bar{\mathbf{x}}_t - \mathbf{x}^\star, \gamma_t^i \mathbf{g}_t^i \right\rangle \overset{(4)}{\leq} -\gamma_t^i \left[ F_i(\bar{\mathbf{x}}_t, \xi_t^i) - F_i(\mathbf{x}_t^i, \xi_t^i) - \frac{L}{2} \left\| \bar{\mathbf{x}}_t - \mathbf{x}_t^i \right\|^2 + F_i(\mathbf{x}_t^i, \xi_t^i) - F_i(\mathbf{x}^\star, \xi_t^i) + \frac{\mu}{2} \left\| \mathbf{x}_t^i - \mathbf{x}^\star \right\|^2 \right]$$

$$\leq -\gamma_t^i \left[ F_i(\bar{\mathbf{x}}_t, \xi_t^i) - F_i(\mathbf{x}^\star, \xi_t^i) \right] + \gamma_t^i L \left\| \bar{\mathbf{x}}_t - \mathbf{x}_t^i \right\|^2 - \frac{\gamma_t^i \mu}{2} \left\| \bar{\mathbf{x}}_t - \mathbf{x}^\star \right\|^2 , \quad (38)$$

where the last inequality used $\left\| \bar{\mathbf{x}}_t - \mathbf{x}^\star \right\|^2 \leq 2 \left\| \bar{\mathbf{x}}_t - \mathbf{x}_t^i \right\|^2 + 2 \left\| \mathbf{x}_t^i - \mathbf{x}^\star \right\|^2$. The last term is essential to get a linear decrease in the first term. For illustration, Equation (36) now changes to

$$\mathbb{E} \left\| \bar{\mathbf{x}}_{t+1} - \mathbf{x}^\star \right\|^2 \leq (1 - \gamma_b \mu) \left\| \bar{\mathbf{x}}_t - \mathbf{x}^\star \right\|^2 - \gamma_b [f(\bar{\mathbf{x}}_t) - f^\star] + 4\gamma_b L \mathbb{E}[R_t] + 4L\gamma_b^2 \sigma_f^2$$

(the constant in front of $\mathbb{E}[R_t]$ increased by a factor of two because of the estimate in (38) and can be controlled by the slightly stronger condition on $\gamma_b$). From there, the proof is very standard and follows exactly the template outlined in e.g. (Stich & Karimireddy, 2020, Proof of Theorem 16) or (Koloskova et al., 2020, Proof of Theorem 2, in particular Lemma 15).

## C Additional details for FedSPS

### C.1 Extending FedSPS to the mini-batch setting

Note that for the sake of simplicity in notation of the proposed method and algorithms, we used a single stochastic sample $\xi_t^i$. However, this can be trivially extended to the mini-batch setting. For a batch $\mathcal{B}_t^i$, the stochastic gradient would become $\frac{1}{|\mathcal{B}_t^i|} \sum_{\xi \in \mathcal{B}_t^i} \nabla F_i(\mathbf{x}_t^i, \xi)$, and the stochastic loss would become $\frac{1}{|\mathcal{B}_t^i|} \sum_{\xi \in \mathcal{B}_t^i} F_i(\mathbf{x}_t^i, \xi)$. We use this mini-batch setting for our practical experiments.

### C.2 Comparison of heterogeneity measures

#### C.2.1 Proof of Proposition 1

*Proof.* We look at the two cases separately as follows:

**Function heterogeneity.** We recall the definitions $\zeta_\star^2 = \frac{1}{n} \sum_{i=1}^n \left\| \nabla f_i(\mathbf{x}^\star) - \nabla f(\mathbf{x}^\star) \right\|_2^2$ and $\sigma_f^2 = \frac{1}{n} \sum_{i=1}^n \left( f_i(\mathbf{x}^\star) - \ell_i^\star \right)$. The global optimal point is denoted by $\mathbf{x}^\star$, and the local optima by $\mathbf{x}_i^\star$. Note that $\nabla f(\mathbf{x}^\star) = 0$, $\nabla f_i(\mathbf{x}_i^\star) = 0$, and we make use of these facts in our proof.

Assuming $L-$smoothness of each $f_i(\mathbf{x})$, we can apply Lemma 7 to each of them with $\mathbf{x} = \mathbf{x}^\star$ to obtain

$$\left\| \nabla f_i(\mathbf{x}^\star) - \nabla f_i(\mathbf{x}_i^\star) \right\|^2 \leq 2L \left( f_i(\mathbf{x}^\star) - f_i^\star \right), \forall i \in [n]. \quad (39)$$

Using this result we can bound $\zeta_\star^2$ from above as follows (noting that $\nabla f(\mathbf{x}^\star) = \mathbf{0}, \nabla f_i(\mathbf{x}_i^\star) = \mathbf{0}$):

$$\zeta_\star^2 = \frac{1}{n} \sum_{i=1}^{n} \|\nabla f_i(\mathbf{x}^\star) - \nabla f(\mathbf{x}^\star)\|_2^2$$

$$= \frac{1}{n} \sum_{i=1}^{n} \|\nabla f_i(\mathbf{x}^\star) - \nabla f_i(\mathbf{x}_i^\star)\|^2$$

$$\overset{(39)}{\leq} \frac{1}{n} \sum_{i=1}^{n} 2L \left( f_i(\mathbf{x}^\star) - f_i^\star \right)$$

$$\overset{(5)}{\leq} \frac{1}{n} \sum_{i=1}^{n} 2L \left( f_i(\mathbf{x}^\star) - \ell_i^\star \right)$$

$$= 2L\sigma_f^2$$

**Gradient variance.** We recall the definitions $\sigma_\star^2 = \frac{1}{n} \sum_{i=1}^{n} \mathbb{E}_{\xi^i} \left\| \nabla F_i(\mathbf{x}^\star, \xi^i) - \nabla f_i(\mathbf{x}^\star) \right\|^2$ and $\sigma_f^2 = \frac{1}{n} \sum_{i=1}^{n} (f_i(\mathbf{x}^\star) - \ell_i^\star)$. The global optimal point is denoted by $\mathbf{x}^\star$, local optimum on client $i$ by $\mathbf{x}_i^\star$ and the local optima of the functions on worker $i$ by $\mathbf{x}_{\xi^i}^\star$. Note that $\nabla f_i(\mathbf{x}_i^\star) = \mathbf{0}$, $\nabla F_i(\mathbf{x}_{\xi^i}^\star, \xi^i) = \mathbf{0}$ and we make use of these facts in our proof.

Assuming $L$ smoothness of each $F_i(\mathbf{x}, \xi^i)$, we can apply Lemma 7 to each function on any client $i$, with $\mathbf{x} = \mathbf{x}^\star$ to obtain

$$\left\| \nabla F_i(\mathbf{x}^\star, \xi^i) - \nabla F_i(\mathbf{x}_{\xi^i}^\star, \xi^i) \right\|^2 \leq 2L \left( F_i(\mathbf{x}^\star, \xi^i) - F_i(\mathbf{x}_{\xi^i}^\star, \xi^i) \right), \forall i \in [n], \xi^i \sim \mathcal{D}_i. \quad (40)$$

Using this result we can bound $\sigma_\star^2$ from above as follows

$$\sigma_\star^2 = \frac{1}{n} \sum_{i=1}^{n} \mathbb{E}_{\xi^i} \left\| \nabla F_i(\mathbf{x}^\star, \xi^i) - \nabla f_i(\mathbf{x}^\star) \right\|^2$$

$$\overset{(20)}{\leq} \frac{1}{n} \sum_{i=1}^{n} \mathbb{E}_{\xi^i} \left\| \nabla F_i(\mathbf{x}^\star, \xi^i) \right\|^2$$

$$= \frac{1}{n} \sum_{i=1}^{n} \left( \mathbb{E}_{\xi^i} \left[ \left\| \nabla F_i(\mathbf{x}^\star, \xi^i) - \nabla F_i(\mathbf{x}_{\xi^i}^\star, \xi^i) \right\|^2 \right] \right)$$

$$\overset{(40)}{\leq} \frac{2L}{n} \sum_{i=1}^{n} \left( \mathbb{E}_{\xi^i} \left[ F_i(\mathbf{x}^\star, \xi^i) - F_i^\star(\mathbf{x}_{\xi^i}^\star, \xi^i) \right] \right)$$

$$\overset{(5)}{\leq} \frac{2L}{n} \sum_{i=1}^{n} \left( \mathbb{E}_{\xi^i} \left[ F_i(\mathbf{x}^\star, \xi^i) - \ell_i^\star \right] \right)$$

$$= \frac{2L}{n} \sum_{i=1}^{n} \left( f_i(\mathbf{x}^\star) - \ell_i^\star \right)$$

$$= 2L\sigma_f^2 \qquad \qquad \square$$

## D    ALTERNATIVE DESIGN CHOICES FOR FEDSPS

### D.1    FEDSPS-NORMALIZED

Drawing ideas from Wang et al. (2021) which employs client and server side normalization to account for any solution bias due to asynchronous local stepsizes (especially for heterogeneous clients), we design FedSPS-Normalized. In the following we provided a detailed description of the algorithm, as well as comparison to our main proposed method FedSPS.

---

**Algorithm 2  FedSPS-Normalized:** Fully locally adaptive FedSPS with normalization to account for heterogeneity as suggested in Wang et al. (2021).

---

**Input:** $\mathbf{x}_0^i = \mathbf{x}_0, \forall i \in [n]$
1: **for** $t = 0, 1, \cdots, T-1$ **do**
2:     **for** each client $i = 1, \cdots, n$ in parallel **do**
3:         sample $\xi_t^i$, compute $\mathbf{g}_t^i := \nabla F_i(\mathbf{x}_t^i, \xi_t^i)$
4:         **FedSPS-Normalized:** $\gamma_t^i = \min \left\{ \frac{F_i(\mathbf{x}_t^i, \xi_t^i) - \ell_i^\star}{c \|\mathbf{g}_t^i\|^2}, \gamma_b \right\}$         ▷ local stochastic Polyak stepsize
5:         **if** $t+1$ is a multiple of $\tau$ **then**
6:             $\mathbf{x}_{t+1}^i = \frac{1}{\frac{1}{n}\sum_{i=1}^n \frac{1}{\nu_i}} \frac{1}{n} \sum_{i=1}^n \frac{1}{\nu_i} \left( \mathbf{x}_t^i - \gamma_t^i \mathbf{g}_t^i \right)$        ▷ aggregation incorporating normalizations
7:             $\nu^i = 0$
8:         **else**
9:             $\mathbf{x}_{t+1}^i = \mathbf{x}_t^i - \gamma_t^i \mathbf{g}_t^i$                   ▷ local step
10:            $\nu^i = \nu^i + \gamma_t^i$                  ▷ client normalization factor
11:         **end if**
12:     **end for**
13: **end for**

---

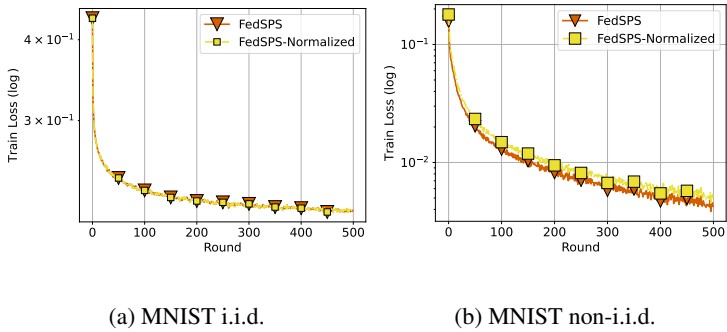

(a) MNIST i.i.d.                            (b) MNIST non-i.i.d.

Figure 6: Comparison of FedSPS and FedSPS-Normalized for the convex case of logistic regression on MNIST dataset (i.i.d. and non-i.i.d.). For the homogeneous case (a) normalization has no effect, while for the heterogeneous case (b) normalization slightly deteriorates performance.

### D.1.1  ALGORITHM

**FedSPS-Normalized.** In FedSPS-Normalized (Algorithm 2), we calculate the client correction factor $\nu^i$ for each client $i \in [n]$ as the sum of local stepsizes (since the last aggregation round). During the server aggregation step, the pseudo-gradients are then normalized using the client and server normalization factors, to remove any bias due to local heterogeneity.

### D.1.2  FEDSPS-NORMALIZED EXPERIMENTAL DETAILS

The implementation is done according to Algorithm 2. All remarks we made about the choice of scaling parameter $c$, upper bound on stepsize $\gamma_b$, and $\ell_i^\star$ in the previous section on FedSPS also remain same here. Figure 6 shows a comparison of FedSPS and FedSPS-Normalized for the convex case of logistic regression on MNIST dataset. For the homogeneous case (a) normalization has no effect—this is as expected since the client and server side correction factors are equal and balance out. We expect to observe some changes for the heterogeneous setting (b), but here the FedSPS-Normalized actually performs very slightly worse than FedSPS. We conclude that normalization does not lead to any significant performance gain, and can intuitively attribute this to the fact the FedSPS stepsizes have low inter-client variance (Figure 7).

### D.2  FEDSPS-GLOBAL

For the sake of comparison with existing locally adaptive FL algorithms such as Local-AMSGrad Chen et al. (2020) and Local-AdaAlter Xie et al. (2019) (both of which perform some form of

---

**Algorithm 3 FedSPS-Global:** FedSPS with global stepsize aggregation

---

**Input:** $\gamma = \gamma_0, \mathbf{x}_0^i = \mathbf{x}_0, \forall i \in [n]$
1: **for** $t = 0, 1, \cdots, T-1$ **do**
2:    **for** each client $i = 1, \cdots, n$ in parallel **do**
3:       sample $\xi_t^i$, and compute stochastic gradient $\mathbf{g}_t^i := \nabla F_i(\mathbf{x}_t^i, \xi_t^i)$
4:       **if** $t+1$ is a multiple of $\tau$ **then**
5:          $\mathbf{x}_{t+1}^i = \frac{1}{n} \sum_{i=1}^n \left( \mathbf{x}_t^i - \gamma \mathbf{g}_t^i \right)$                           ▷ communication round
6:         **FedSPS-Global:** $\gamma = \frac{1}{n} \sum_{i=1}^n \min \left\{ \frac{F_i(\mathbf{x}_t^i, \xi_t^i) - \ell_i^\star}{c \|\mathbf{g}_t^i\|^2}, \gamma_b \right\}$        ▷ global stepsize
7:       **else**
8:          $\mathbf{x}_{t+1}^i = \mathbf{x}_t^i - \gamma \mathbf{g}_t^i$                                      ▷ local step
9:       **end if**
10:    **end for**
11: **end for**

---

stepsize aggregation on the server), we introduce a synchronous stepsize version of our algorithm called FedSPS-Global.

### D.2.1 ALGORITHM

**FedSPS-Global.** In FedSPS-Global (Algorithm 3), the stepsize is constant across all clients and also for local steps for a particular client, for a particular round $t$. There can be several choices for the aggregation formula for the "global" stepsize—we use a simple average of the local SPS stepsizes as given below

$$\gamma = \frac{1}{n} \sum_{i=1}^n \min \left\{ \frac{F_i(\mathbf{x}_t^i, \xi_t^i) - \ell_i^\star}{c \left\| \nabla F(\mathbf{x}_t^i, \xi_t^i) \right\|^2}, \gamma_b \right\} . \tag{41}$$

We provide a theoretical justification of this choice of aggregation formula, and compare it with other possible choices in Appendix D.2.2. As it is apparent from (41), we need to use stale quantities from the last round, for calculating the stepsize for the current round. This is justified by the empirical observation that stepsizes do not vary too much between consecutive rounds, and is a reasonable assumption that has also been made in previous work on adaptive distributed optimization Xie et al. (2019). Due to the staleness in update of the stepsize in this method, we need to provide a starting stepsize $\gamma_0$ to the algorithm, and this stepsize is used in all clients till the first communication round happens. FedSPS-Global can be thought of more as a heuristic method using cheap stepsize computations, and offering similar empirical performance as FedSPS.

### D.2.2 CHOICE OF AGGREGATION FORMULA FOR FEDSPS-GLOBAL

We can have various choices for aggregating the local stepsizes to calculate the global stepsize. The first obvious choice would be to perform a simple average of the local stepsizes across all clients. This is given by the aggregation formula $\gamma^{(a)} = \frac{1}{n} \sum_{i=1}^n \min \left\{ \frac{f_i(\mathbf{x}_t^i) - \ell_i^\star}{c \|\mathbf{g}_t^i\|^2}, \gamma_b \right\}$ which is used in our proposed method (Algorithm 3). Two other plausible choices of aggregation formula could be $\gamma^{(b)} = \min \left\{ \frac{\frac{1}{n} \sum_{i=1}^n [f_i(\mathbf{x}_t^i) - \ell_i^\star]}{c \frac{1}{n} \sum_{i=1}^n \|\mathbf{g}_t^i\|^2}, \gamma_b \right\}$ and $\gamma^{(c)} = \min \left\{ \frac{\frac{1}{n} \sum_{i=1}^n [f_i(\mathbf{x}_t^i) - \ell_i^\star]}{c \left\| \frac{1}{n} \sum_{i=1}^n \mathbf{g}_t^i \right\|^2}, \gamma_b \right\}$. Among these choices, $\gamma^{(c)}$ represents the "correct" SPS stepsize if we follow the original definition and replace batches with clients in the distributed setup. In the following Proposition we show theoretically that $\gamma^{(b)} < \min \left\{ \gamma^{(a)}, \gamma^{(c)} \right\}$. Experimentally we find the simple averaging of local stepsizes i.e., $\gamma^{(a)}$ to work best, followed closely by $\gamma^{(a)}$, while the "correct" SPS stepsize $\gamma^{(c)}$ explodes in practice. Therefore, we choose $\gamma^{(a)}$ as our aggregation formula and this has some added advantages for the associated proofs. We feel that a reason behind the good performance of FedSPS-Global is the low inter-client and intra-client variance of the stepsizes explained in Section 6—this is the reason behind why simple averaging of the local stepsizes work.

**Proposition 2** (Global SPS stepsize aggregation formula). *Using the definitions of the three aggregation formulae for synchronous SPS stepsizes $\gamma^{(a)}$, $\gamma^{(b)}$, and $\gamma^{(c)}$, as defined above in Section D.2.2, we have the following inequalities*

1. $\gamma^{(c)} \geq \gamma^{(b)}$.

2. $\gamma^{(a)} \geq \gamma^{(b)}$.

*combining which we get* $\gamma^{(b)} < \min\left\{\gamma^{(a)}, \gamma^{(c)}\right\}$.

*Proof.* We look at the two cases separately as follows:

1. From Lemma 10 it is easy to observe that

$$\left\|\sum_{i=1}^{n} \mathbf{g}_t^i\right\|^2 \overset{(17)}{\leq} n \sum_{i=1}^{n} \left\|\mathbf{g}_t^i\right\|^2$$

which can be rearranged as

$$\frac{\sum_{i=1}^{n}[f_i(\mathbf{x}_t^i) - \ell_i^\star]}{\left\|\frac{1}{n}\sum_{i=1}^{n}\mathbf{g}_t^i\right\|^2} \geq \frac{\sum_{i=1}^{n}[f_i(\mathbf{x}_t^i) - \ell_i^\star]}{\frac{1}{n}\sum_{i=1}^{n}\left\|\mathbf{g}_t^i\right\|^2}$$

The required statement follows trivially from the above inequality.

2. From Chebyshev's inequality we have

$$\frac{1}{n}\sum_{i=1}^{n}\frac{f_i(\mathbf{x}_t^i) - \ell_i^\star}{\left\|\mathbf{g}_t^i\right\|^2} \geq \frac{\sum_{i=1}^{n}[f_i(\mathbf{x}_t^i) - \ell_i^\star]}{n} \cdot \frac{1}{n}\sum_{i=1}^{n}\frac{1}{\left\|\mathbf{g}_t^i\right\|^2}$$

From AM-HM inequality we obtain

$$\frac{\sum_{i=1}^{n}\frac{1}{\left\|\mathbf{g}_t^i\right\|^2}}{n} \geq \frac{n}{\sum_{i=1}^{n}\left\|\mathbf{g}_t^i\right\|^2}$$

Plugging in this into the above we get

$$\frac{1}{n}\sum_{i=1}^{n}\frac{f_i(\mathbf{x}_t^i) - \ell_i^\star}{\left\|\mathbf{g}_t^i\right\|^2} \geq \frac{\frac{1}{n}\sum_{i=1}^{n}[f_i(\mathbf{x}_t^i) - \ell_i^\star]}{\frac{1}{n}\sum_{i=1}^{n}\left\|\mathbf{g}_t^i\right\|^2}$$

The required statement follows trivially from the above inequality.

□

### D.2.3 FEDSPS-GLOBAL EXPERIMENTAL DETAILS

The implementation is done according to Algorithm 3[3]. All remarks we made about the choice of scaling parameter $c$, upper bound on stepsize $\gamma_b$, and $\ell_i^\star$ in the previous section on FedSPS also remain same here. As mentioned before, this method needs an extra hyperparameter $\gamma_0$, that is the stepsize across all clients until the first communication round. Empirically we found that setting $\gamma_0 = \gamma_0^0$, i.e. the local SPS stepsize for client 0 at iteration 0, works quite well in practice. This is explained by the fact that the SPS stepsizes have low inter-client and intra-client variance (Figure 7). Experimentally we find that FedSPS-Global converges almost identically to the locally adaptive FedSPS (Figure 4 and Figure 9).

## E  EXTENSION OF FEDSPS TO NON-CONVEX SETTING

We outline here some theoretical results of extending FedSPS to the non-convex setting. Our non-convex results have the limitation of requiring small stepsizes, but we do not need the additional strong assumption of bounded stochastic gradients used in previous work on adaptive federated optimization.

---

[3]Note that for any experiment in the Appendix that involves FedSPS-Global, we refer to FedSPS (from the main paper) as FedSPS-Local in the legends of plots, for the sake of clarity between the two approaches.

### E.1 CONVERGENCE ANALYSIS FOR NON-CONVEX FUNCTIONS

We now discuss the convergence of FedSPS on non-convex functions. Unfortunately, it is required to impose additional assumptions in this section. This is mainly due that the Polyak stepsize was designed for convex objectives and additional assumptions are needed in the non-convex setting.

**Assumption 3** (Bounded variance). *Each function $f_i \colon \mathbb{R}^d \to \mathbb{R}$, $i \in [n]$ has stochastic gradient with bounded local variance, that is, for all $\mathbf{x} \in \mathbb{R}^d$,*

$$\mathbb{E}_{\xi \sim \mathcal{D}_i} \|\nabla F_i(\mathbf{x}, \xi) - \nabla f_i(\mathbf{x})\|_2^2 \le \sigma^2 . \tag{42}$$

*Moreover, global variance of the loss function on each client is bounded, that is, for all $\mathbf{x} \in \mathbb{R}^d$,*

$$\frac{1}{n} \sum_{i=1}^{n} \|\nabla f_i(\mathbf{x}) - \nabla f(\mathbf{x})\|_2^2 \le \zeta^2 . \tag{43}$$

This assumption is frequently used in the analysis of FedAvg Koloskova et al. (2020), as well as adaptive federated methods Reddi et al. (2021); Wang et al. (2022a). The local variance denotes randomness in stochastic gradients of clients, while the global variance represents heterogeneity between clients. Note that $\zeta = 0$ corresponds to the i.i.d. setting. We further note that Equation (43) corresponds to the variance Assumption in Loizou et al. (2021) (their Equation (8) with $\rho = 1$, $\delta = \zeta^2$) that they also required for the analysis in the non-convex setting.

The following theorem applies to FedSPS and FedAvg with small stepsizes:

**Theorem 18** (Convergence of FedSPS for non-convex case). *Under Assumptions 1 and 3, if $\gamma_b < \min\left\{\frac{1}{2cL}, \frac{1}{25L\tau}\right\}$ then after $T$ steps, the iterates of FedSPS and FedAvg satisfy*

$$\Phi_T = \mathcal{O}\left(\frac{F_0}{\gamma_b T} + \gamma_b \frac{L\sigma^2}{n} + \gamma_b^2 L^2 \tau(\sigma^2 + \tau\zeta^2)\right) \tag{44}$$

*where $\Phi_T := \min_{0 \le t \le T-1} \mathbb{E} \|\nabla f(\bar{\mathbf{x}}_t)\|^2$ and $F_0 := f(\bar{\mathbf{x}}_0) - f^\star$.*

The proof of this theorem again relies on the assumption that the stepsize is very small, but otherwise follows the template of earlier work Li et al. (2019); Koloskova et al. (2020) and precisely recovers their result. For the sake of completeness we still add a proof in the Appendix, but do not claim novelty here. The theorem states that when using a constant stepsize, the algorithms reach a neighborhood of the solution, where the neighbourhood is dependent on the local variance $\sigma$ and global variance $\zeta$, and for the i.i.d. case of $\zeta = 0$ the neighbourhood is smaller and less dependent on number of local steps $\tau$. By choosing an appropriately small $\gamma_b$, any arbitrary target accuracy $\epsilon$ can be reached.

A limitation of our result is that it only applies to the small stepsize regime, when the adaptivity is governed by the stepsize bound on $\gamma_b$. However, when comparing to other theoretical work on adaptive federated learning algorithms we observe that related work has similar (or stronger) limitations, as e.g. both Wang et al. (2022a) (FedAMS) and Reddi et al. (2021) (FedAdam) require an uniform bound on the stochastic gradients (we do not need) and also require effective stepsizes smaller than $\frac{1}{\tau L}$ similar to our case.

#### E.1.1 PROOF OF THEOREM 18

**Small Stepsize.** We start by the observation that it must hold $\gamma_t^i = \gamma_b$ for all iterations $t$ and clients $i$. This is due to our strong assumption on the small stepsizes.

**Bound on $R_t$.** Similar as in the convex case, we need a bound on $\mathbb{E}[R_t]$. We note that when deriving the bound in Equation (27) we did not use any convexity assumption, so this bound also holds for arbitrary smooth functions:

$$R_t \le \frac{32\gamma_b^2 \tau}{n} \sum_{i=1}^{n} \sum_{j=(t-1)-k(t)}^{t-1} \left\|F_i(\bar{\mathbf{x}}_j, \xi_j^i)\right\|^2 .$$

and consequently (after taking expectation) combined with Assumption 3:

$$\mathbb{E}\, R_t \le 32\gamma_b^2\tau \sum_{j=(t-1)-k(t)}^{t-1} [\mathbb{E}\,\|\nabla f(\bar{\mathbf{x}}_j)\|^2 + \sigma^2/\tau + \zeta^2]$$

$$\le \frac{1}{16L^2\tau} \sum_{j=(t-1)-k(t)}^{t-1} \mathbb{E}\,\|\nabla f(\bar{\mathbf{x}}_j)\|^2 + 32\gamma_b^2\tau^2[\sigma^2/\tau + \zeta^2]\,, \tag{45}$$

where the last estimate followed by $\gamma_b \le \frac{1}{25\tau L}$.

**Decrease.** We study the (virtual) average $\bar{\mathbf{x}}_t$. By smoothness and definition of $\bar{\mathbf{x}}_{t+1}$ we have:

$$f(\bar{\mathbf{x}}_{t+1}) \le f(\bar{\mathbf{x}}_t) - \frac{\gamma_b}{n}\sum_{i=1}^n \langle \nabla f(\bar{\mathbf{x}}_t), \mathbf{g}_t^i \rangle + \frac{\gamma_b^2 L}{2}\left\|\frac{1}{n}\sum_{i=1}^n \mathbf{g}_t^i\right\|^2. \tag{46}$$

The scalar product can be estimated as follows:

$$-\frac{1}{n}\sum_{i=1}^n \langle \nabla f(\bar{\mathbf{x}}_t), \mathbf{g}_t^i \rangle = -\frac{1}{n}\sum_{i=1}^n \left( \langle \nabla f(\bar{\mathbf{x}}_t), \nabla F_i(\bar{\mathbf{x}}_t, \xi_t^i) \rangle + \langle \nabla f(\bar{\mathbf{x}}_t), \nabla F_i(\mathbf{x}_t^i, \xi_t^i) - \nabla F_i(\bar{\mathbf{x}}_t^i, \xi_t^i) \rangle \right)$$

$$\le -\frac{1}{n}\sum_{i=1}^n \langle \nabla f(\bar{\mathbf{x}}_t), \nabla F_i(\bar{\mathbf{x}}_t, \xi_t^i) \rangle + \frac{1}{2}\|\nabla f(\bar{\mathbf{x}}_t)\|^2 + \frac{1}{2n}\sum_{i=1}^n \|\nabla F_i(\mathbf{x}_t^i, \xi_t^i) - \nabla F_i(\bar{\mathbf{x}}_t^i, \xi_t^i)\|^2$$

$$= -\frac{1}{n}\sum_{i=1}^n \langle \nabla f(\bar{\mathbf{x}}_t), \nabla F_i(\bar{\mathbf{x}}_t, \xi_t^i) \rangle + \frac{1}{2}\|\nabla f(\bar{\mathbf{x}}_t)\|^2 + \frac{1}{2}L^2 R_t\,.$$

and after taking expectation:

$$-\mathbb{E}\left[\frac{1}{n}\sum_{i=1}^n \langle \nabla f(\bar{\mathbf{x}}_t), \mathbf{g}_t^i \rangle\right] \le -\frac{1}{2}\|\nabla f(\bar{\mathbf{x}}_t)\|^2 + \frac{1}{2}L^2 R_t$$

We now also use smoothness to estimate the last term in (46):

$$\mathbb{E}\left\|\frac{1}{n}\sum_{i=1}^n \mathbf{g}_t^i\right\|^2 = \mathbb{E}\left\|\frac{1}{n}\sum_{i=1}^n \nabla F_i(\mathbf{x}_t^i, \xi_t^i) - \nabla f_i(\mathbf{x}_t^i)\right\|^2 + \left\|\frac{1}{n}\sum_{i=1}^n \nabla f_i(\mathbf{x}_t^i)\right\|^2$$

$$\le \frac{\sigma^2}{n} + 2\|\nabla f(\bar{\mathbf{x}}_t)\|^2 + 2\left\|\frac{1}{n}\sum_{i=1}^n \nabla f_i(\mathbf{x}_t^i) - \nabla f_i(\bar{\mathbf{x}}_t)\right\|^2$$

$$\le \frac{\sigma^2}{n} + 2\|\nabla f(\bar{\mathbf{x}}_t)\|^2 + 2L^2 R_t\,.$$

By combining these estimates, and using $\gamma_b \le \frac{1}{10L}$, we arrive at:

$$\mathbb{E}\,f(\bar{\mathbf{x}}_{t+1}) \le f(\bar{\mathbf{x}}_t) - \frac{\gamma_b}{2}\|\nabla f(\bar{\mathbf{x}}_t)\|^2 + \gamma_b L^2 R_t + \frac{\gamma_b^2 L}{2}\left(2\|\nabla f(\bar{\mathbf{x}}_t)\|^2 + \frac{\sigma^2}{n} + 2L^2 R_t\right)$$

$$\le f(\bar{\mathbf{x}}_t) - \frac{\gamma_b}{4}\|\nabla f(\bar{\mathbf{x}}_t)\|^2 + 2\gamma_b L^2 R_t + \frac{\gamma_b^2 L\sigma^2}{2n}\,.$$

We now re-arrange and plug in the estimate of $R_t$ from (45):

$$\frac{\gamma_b}{4}\mathbb{E}\,\|\nabla f(\bar{\mathbf{x}}_t)\|^2 \le \mathbb{E}\,f(\bar{\mathbf{x}}_t) - f(\bar{\mathbf{x}}_{t+1}) + \frac{\gamma_b^2 L\sigma^2}{2n} + \frac{\gamma_b}{8\tau}\sum_{j=(t-1)-k(t)}^{t-1} \mathbb{E}\,\|\nabla f(\bar{\mathbf{x}}_j)\|^2 + 64\gamma_b^3\tau^2 L^2[\sigma^2/\tau + \zeta^2]$$

and after summing from $t = 0$ to $T-1$ and dividing by $T$:

$$\frac{\gamma_b}{4T}\sum_{t=0}^{T-1} \mathbb{E}\,\|\nabla f(\bar{\mathbf{x}}_t)\|^2 \le \frac{1}{T}\sum_{t=0}^{T-1} (\mathbb{E}\,f(\bar{\mathbf{x}}_t) - f(\bar{\mathbf{x}}_{t+1})) + \frac{\gamma_b^2 L\sigma^2}{2n} + 64\gamma_b^3\tau^2 L^2[\sigma^2/\tau + \zeta^2]$$

$$+ \frac{\gamma_b}{8T}\sum_{t=0}^{T-1} \mathbb{E}\,\|\nabla f(\bar{\mathbf{x}}_t)\|^2$$

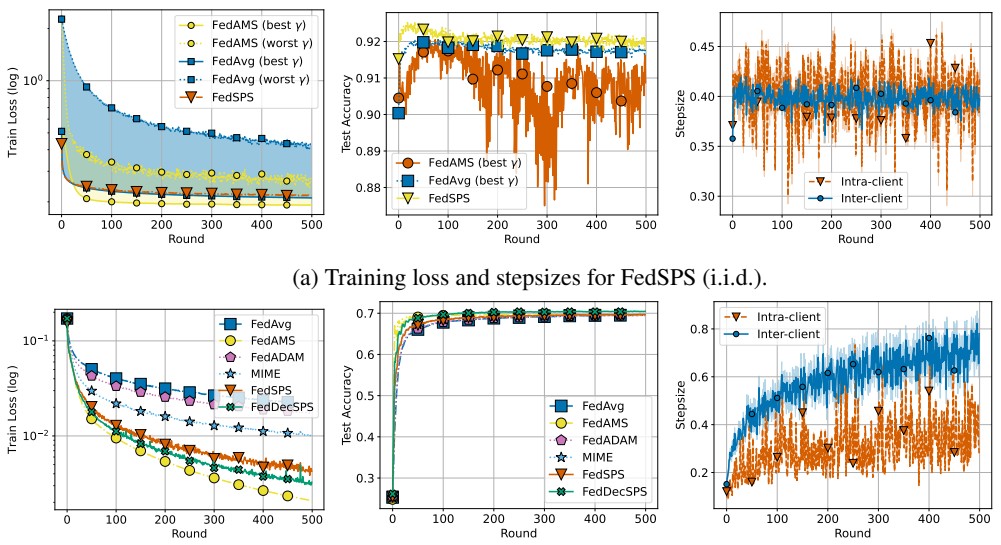

(a) Training loss and stepsizes for FedSPS (i.i.d.).

(b) Training loss, test accuracy and stepsizes for FedSPS (non-i.i.d.).

Figure 7: Visualization of FedSPS stepsize statistics for convex case of logistic regression on MNIST dataset (i.i.d. and non-i.i.d.). Left column represents training loss, middle column test accuracy, and right column stepsize plots. "intra-client" refers to selecting any one client and plotting the mean and standard deviation of stepsizes for that client across $\tau$ local steps, for a particular communication round. "inter-client" refers plotting the mean and standard deviation of stepsizes across all clients, for a particular communication round.

implying that

$$\frac{1}{8T}\sum_{t=0}^{T-1}\mathbb{E}\left\|\nabla f(\bar{\mathbf{x}}_t)\right\|^2 \leq \frac{1}{T\gamma_b}(f(\bar{\mathbf{x}}_0) - f^\star) + \frac{\gamma_b L\sigma^2}{2n} + 64\gamma_b^2\tau^2 L^2(\sigma^2/\tau + \zeta^2)\,.$$

# F  ADDITIONAL EXPERIMENTS

## F.1  VISUALIZATION OF FEDSPS STEPSIZE STATISTICS

Intuitively it may seem at first glance that using fully locally adaptive stepsizes can lead to poor convergence due to different stepsizes on each client. However, as we have already seen in our theory and well as verified in experiments, that this is not the case—our fully locally adaptive FedSPS indeed converges. In this remark, we try to shed further light into the convergence of FedSPS by looking at the stepsize statistics across clients. Figure 7 plots the "intra-client" and "inter-client" stepsize plots for i.i.d. and non-i.i.d. experiments. "intra-client" visualizes the variance in stepsizes for a particular client across different local steps. "inter-client" visualizes the the variance in stepsizes across all clients. We can notice that both these variances are small, explaining the good convergence behaviour of FedSPS.

## F.2  EFFECT OF VARYING $c$ ON CONVERGENCE

We provide additional experiments on the effect of varying the FedSPS scale parameter $c$ on convergence, for different number of local steps $\tau \in \{10, 20, 50, 100\}$ (convex case logistic regression on i.i.d. MNIST dataset without client sampling) in Figure 8. Similarly as before, we observe that smaller $c$ results in better convergence, and any $c \in \{0.01, 0.1, 0.5\}$ works well. Moreover, the effect of varying $c$ on the convergence is same for all $\tau$ clarifying that in practice there is no square law relation between $c$ and $\tau$, unlike as suggested by the theory.

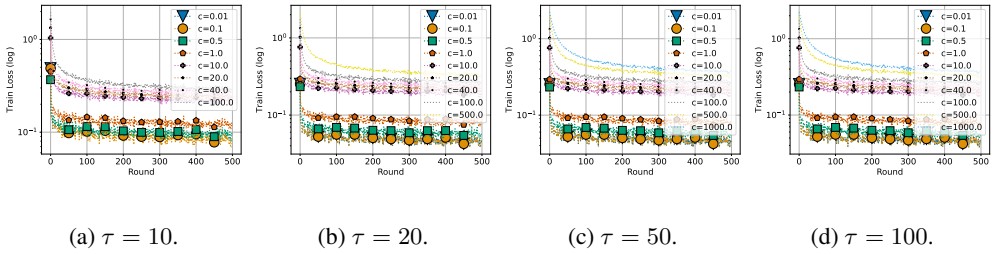

(a) $\tau = 10$.     (b) $\tau = 20$.     (c) $\tau = 50$.     (d) $\tau = 100$.

Figure 8: Additional experiment on the effect of varying $c$ on convergence of FedSPS for different values of $\tau$. In practice, there is no square law relation between $c$ and $\tau$, and any small value of $c \leq 0.5$ works well.

Table 1: Hyperparameters for FedAvg and FedAMS.

| Dataset | Architecture | $\eta_l$ | $\eta$ | $\epsilon$ |
|---------|-------------|----------|--------|------------|
| LIBSVM | Logistic Regression | 1.0 | 1 | 0.01 |
| MNIST | Logistic Regression LeNet | 0.1 | 1 | 0.001 |

## F.3   ADDITIONAL CONVEX EXPERIMENTS

Additional convex experiments for the rest of the LIBSVM datasets (i.i.d.) have been shown in Figure 9. The first column represents the training loss and the second column the test accuracy. The third column represents the FedSPS stepsize statistics as described before in Section F.1. We can make similar observations as before in the main paper—the proposed FedSPS methods (without tuning) converge as well as or slightly better than FedAvg and FedAdam with the best tuned hyperparameters. The stepsize plots again highlight the low inter-client and intra-client stepsize variance.

## F.4   HYPERPARAMETERS TUNING FOR FEDAVG AND FEDAMS

Here we provide more details on hyperparameters for each dataset and model needed by FedAvg and FedAMS. We perform a grid search for the client learning rate $\eta_l \in \{0.0001, 0.001, 0.01, 0.1, 1.0\}$, and server learning rate $\eta \in \{0.001, 0.01, 0.1, 1.0\}$. Moreover, for FedAMS we also choose $\beta_1 = 0.9$, $\beta_2 = 0.99$, and the max stabilization factor $\epsilon \in \{10^{-8}, 10^{-4}, 10^{-3}, 10^{-2}, 10^{-1}, 10^{0}\}$. The grid search leads to the following set of optimal hyperparameters presented in Table 1.

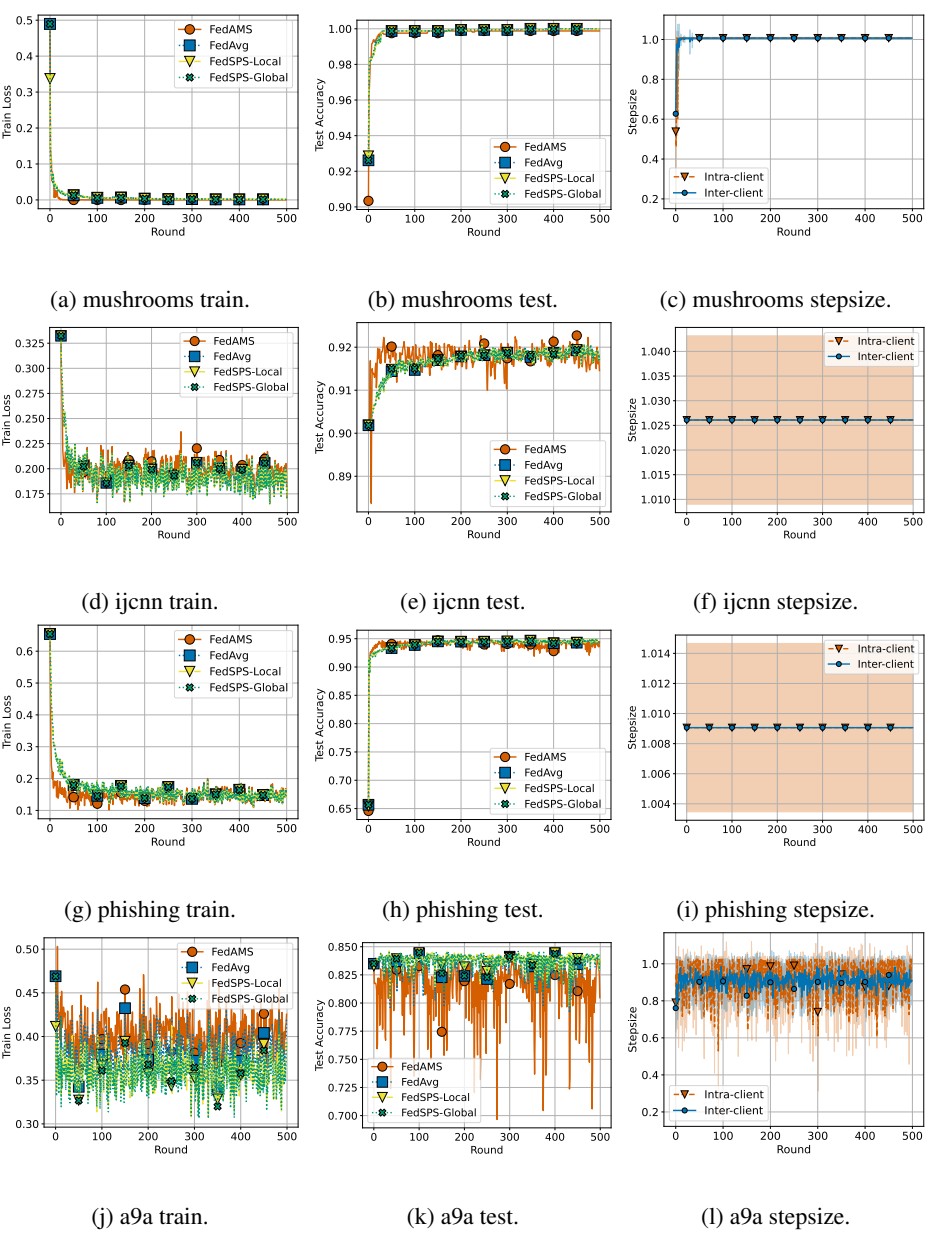

Figure 9: Additional convex experiments for the LIBSVM datasets (i.i.d.). As mentioned in Section D.2.3, FedSPS (from the main paper) is referred to as FedSPS-Local here in the legends, to distinguish it clearly from FedSPS-Global.

