# OpenReview forum: "Locally Adaptive Federated Learning"
_ICLR.cc/2024/Conference — Submitted to ICLR 2024_

### Official Review · Reviewer_e5wf · 2023-10-29

**Soundness:** 2 fair
**Presentation:** 2 fair
**Contribution:** 2 fair
**Rating:** 5
**Confidence:** 3

**Summary:**

The paper makes substantial use of the Polyak stepsize, a well-known technique in optimization for determining the learning rate based on the function values and gradients. The paper provides an example demonstrating how local adaptivity using Polyak stepsizes can improve convergence in optimization problems. It illustrates a scenario where the use of locally adaptive distributed Gradient Descent with Polyak stepsizes results in a near-constant iteration complexity, which is significantly better than using mini-batch Gradient Descent with a constant stepsize. The paper also delves into the convergence analysis on strongly convex functions. The algorithm is designed to be fully locally adaptive, catering to the needs of each client function in the federated learning setting.

**Strengths:**

1. Originality: The paper introduces a approach to federated learning, addressing the limitations of existing stepzise tuning methods and providing a solution that leverages local geometric information.
2. Quality: The authors attempt to build a theoretical foundation for their proposed algorithms, analyzing their convergence in various settings.

**Weaknesses:**

1. The connection to the Polyak stepsize and the rationale behind the specific choices of (\gamma_1) and (\gamma_2) in Example 1 could be clarified by referring to the definition in Loizou et al. 2021.

2. The choice of a noise standard deviation (sd) of 10 in Figure 1's caption requires clarification, especially given the observation that SPS does not seem to converge.

3. The paper should provide a clear definition of ( f^* ) in Eq. 5, addressing whether it refers to the global minimum of the finite sum or average sum of each f_i. They are essentially different.

4. It would be very hard to parse the sentence that \sigma_f^2 is stronger than (zeta_*, sigma_*) but weaker than (zeta, sigma), and it is very hard to connect that to the inequalities in Proposition 1.

5. The paper addressed the apparent need for hyperparameter tuning in both convex and non-convex experiments for FedSPS, especially given the gap between the worst and best performance from FedSPS and the gap from FedAMS, two of which are very comparable. FedAMS shows better performance in the experiments. So I don't see the remarkable improvement via the proposed method.

6. The paper claims to compare the proposed methods with FedADAM, but this comparison is not present in the paper. Including this missing comparison, more extensive comparisons with other non-iid FL papers, would strengthen the paper:

pFedMe: Personalized Federated Learning with Moreau Envelopes Dinh et al., 2020
PerFedAvg: Personalized Federated Learning with Theoretical Guarantees: A Model-Agnostic Meta-Learning Approach Fallah et al., 2020
APFL: Adaptive Personalized Federated Learning Deng et al., 2020
Ditto: Fair and Robust Federated Learning Through Personalization Li et al., 2022

**Questions:**

1.    - Can you provide more details on how the Polyak stepsize is connected to the choices of (gamma_1) and (gamma_2) in Example 1?
   - How do the specific choices of (gamma_1) and (gamma_2) in Example 1 relate to the definition of Polyak stepsize provided in Loizou et al., 2021?
   - Could you elaborate on the rationale behind selecting these particular values for (gamma_1) and (gamma_2)?

2.    - Why was a noise standard deviation (sd) of 10 chosen for the experiments depicted in Figure 1?
   - Given that SPS does not seem to converge in this scenario, could you explain how the chosen noise level impacts the convergence of SPS?

---

> ### Author Response · Authors · 2023-11-20
>
> We thank the reviewer for their evaluation of our manuscript. We did our best to understand and respond to their constructive criticisms, and provide detailed answers below:
>
> - (W1) For convex and strongly convex functions (such as the ones in Example 1), the optimal local stepsizes can be calculated using the Polyak stepsize (PS). For the above example, we have $f_1(x) = \frac{a}{2}x^2$ and corresponding $\gamma_1^\star = \frac{\frac{a}{2}x_0^2 - 0}{\left(ax_0\right)^2} = \frac{1}{2a}$. Similarly, $f_2(x) = \frac{1}{2}x^2$ and corresponding $\gamma_2^\star = \frac{\frac{1}{2}x_0^2 - 0}{\left(x_0\right)^2} = \frac{1}{2}$. Since this analytical Example considers deterministic gradient descent (GD), we use the deterministic version of the Polyak stepsizie. For Figure 1, we consider the stochastic scenario (SGD) and hence use the SPS stepsize from Loizou et al., 2021.
> - (W2) For Figure 1, SPS does converge. We had set an extremely small threshold error ($\epsilon = 10^{-6}$), beyond which we stop the algorithm. SPS achieves the desired error faster than the rest of the methods, as is obvious from Figure 1. The choice of $\sigma = 10$ to introduce stochasticity is arbitrary, and should not change the observation. Nevertheless, we would add some additional plots for different values of $\sigma$ in our potential camera-ready version, for the sake of completeness.
> - (W3) $f^{\star}$ has been defined in the Introduction section at Equation (1), and refers to the global minimum of the finite sum. In the same paragraph, we also clarify that the global minimum $x^{\star}$ and the local minima $x_i^{\star}$ can be different.
> - (W4) Proposition 1 makes it clear that $\sigma_f$ is a stronger assumption than ($\zeta_{\star}, \sigma_{\star}$). This is because any existing convergence rate for FedAvg using ($\zeta_{\star}, \sigma_{\star}$), will also hold for $\sigma_f$ due to Proposition 1. Moreover, it should also be obvious that uniformly bounded noise ($\zeta, \sigma$) is a stronger assumption than bounded noise at optimum ($\zeta_{\star}, \sigma_{\star}$). We would try to rephrase this comment in the more understandable way in our revised version.
> - (W5) We would like to kindly draw the reviewer’s attention to the non convex experiments (Figures 4 and 5), where our proposed FedSPS in fact performs better than FedAMS.
> - (W6) We do compare our method to FedADAM as claimed in our manuscript, and refer the reviewer to Figure 3(c). For the other experiments we chose to include the comparison to only FedAMS for the sake of brevity, since FedAMS [1] was already shown to empirically beat FedADAM [2]. Nonetheless, we will make sure to add the comparison to FedADAM in the remaining plots for the potential camera-ready version of our paper, to aid better understanding.
> - (Q1) Please refer to the response to (W1) above.
> - (Q2) Please refer to the response to (W2) above.
>
>
> We believe that all the questions raised by reviewer e5wf have been satisfactorily answered in our response above. We shall take care to incorporate the clarifications and changes outlined above in the potential camera-ready version of our paper. If you feel there are no serious flaws in the theoretical claims kindly consider increasing the "Soundness" score. If you also agree that we managed to address all issues raised, please consider increasing your "Contribution" score, as well as the overall "Rating". If you believe this is not the case, please let us know so that we have a chance to respond further.
>
> **References:**
> - [1] Wang, Yujia, Lu Lin, and Jinghui Chen. "Communication-efficient adaptive federated learning." International Conference on Machine Learning. PMLR, 2022.
> - [2] Reddi, S. J., Charles, Z., Zaheer, M., Garrett, Z., Rush, K., Konecny, J., Kumar, S., and McMahan, H. B. Adaptive federated optimization. In International Conference on Learning Representations, 2021.

---

### Official Review · Reviewer_vVZ6 · 2023-10-30

**Soundness:** 3 good
**Presentation:** 3 good
**Contribution:** 2 fair
**Rating:** 6
**Confidence:** 4

**Summary:**

The paper proposes FedSPS, which is an extension of the SPS (stochastic Polyak stepsize) framework [Loizou et al., 2021] to the federated learning setting. A variant of FedDecSPS with decreasing stepsize is also proposed. Convergence with convex loss is provided. Experiments are conducted to show the advantage of FedSPS compared with related FL methods with adaptive learning rates.

**Strengths:**

The method is simple and seems effective in some cases based on the experimental results. FL is an important and hot topic which would be insteresting to the ICLR audience.

The experiments compared with many baseline methods with adaptive learning rates.

**Weaknesses:**

1. Algorithm design: in my understanding, FedSPS is mainly an FL version of SPS [Loizou et al., 2021]. This extension is rather standard and the algorithmic novelty is not particularly strong.

2. Theory: the theoretical analysis combines the techniques of SPS with standard FL convergence proof, and only studied convex loss functions. Many results in the paper require a very small learning rate upper bound $\gamma_b$ (typically for non-iid clients which is common in FL), which significantly limits the 'adaptivity' of FedSPS and the proposed method becomes FedAvg approximately.

3. Experiments:

(1) The presented numerical results do not fully justify the benefit of Polyak learning rates, and some results need more justification. In Figure 2(b), when $\gamma_b=1$, the stepsizes are around 0.87 and very stable through iterations. It never reached 1. That means we are using the Polyak stepsize all the time. As a result, $\gamma_b=5$ should give exactly the same training trajectory as $\gamma_b=1$, right? This is because neither of them trigger the upper bound $\gamma_b$. But in the figure, they are very different.

(2) Also, from Figure 2(a), $\gamma_b=1$ performs the best. From 2(b), the effective stepsizes of $\gamma_b=1$ is very stable. To a large extent, I would say that this is almost a constant learning rate without adaptivity. In contrast, $\gamma_b=5$ really brings adaptive stepsizes because the y-axis jumps a lot through iterations. So it is not very clear to me how 'adaptivity' helps the FL training.

(3) In Figure 4(b), why does FedAdam perform so poorly (almost diverging)? Non-iid MNIST is a standard setting and an easy task. In the original paper of FedAdam and FedAMS there are also MNIST experiments and their methods performed well. This result does not seem very plausible.

2. While the paper claimed that FedSPS needs little parameter tuning, I don't think this necessarily holds in practice. For adaptive optimization based methods (with Adam-type updates), in most cases the default $\beta_1$, $\beta_2$ and $\epsilon$ values already achieve very promising performance, so for FedAdam or LocalAMS we essentially only need to tune the global and local learning rates. Furthermore, in fact, usually setting the global learning rate to 1 performs well. And for FedSPS, if we want, we can also add a global learning rate to (slightly) improve the performance. Moreover, The variant FedDecSPS has two tuning parameters, $\gamma_b$ and $c$.

Therefore, in general, I think the proposed method would require the same amount of parameter tuning as other adaptive FL methods.

**Questions:**

Questions and suggestions:

1. The proposed method is called adaptive FL, but it is different from the commonly noted adaptive methods (e.g., Adam, AMSGrad, etc.) which uses first and second order momentums. FedSPS is more like FedSGD with adaptive stepsizes. For better clarity on the contributions, I suggest that the title could follow [Stochastic Polyak Step-size for SGD: An Adaptive Learning Rate for Fast Convergence, AISTATS 2021] and include 'Polyak Step-size' and 'SGD'.

2. How does your analysis extend to the partial participation setting? I suggest adding a brief statement on this for clarity. For FedAdam,
[Analysis of Error Feedback in Federated Non-convex Optimization with Biased Compression: Fast Convergence and Partial Participation, ICML 2023] might be a relevant but missing reference.

I general, I think this is a borderline paper and more justification is needed. I will be happy to raise the score if my questions are answered well.

---

> ### Author Response · Authors · 2023-11-20
>
> We thank the reviewer for their very thorough evaluation of our manuscript, positive remarks, as well as constructive criticism. We provide detailed answers to the concerns below:
> - (W1) **Algorithm:** As explained in Remark 2, we experimented with various other algorithm design choices for incorporating SPS in FedAvg, such as FedSPS-Global and FedSPS-Normalized which are more complicated than our proposed FedSPS but did not offer any empirical benefits---hence our choice.
> - (W2) **Theory:** The primary theoretical challenge in analysis of FedSPS was extending the error-feedback framework (that originally works for equal stepsizes) to work for fully un-coordinated local stepsizes, and we did this for the first time in our work.
> - (W3) **Experiments:** Following are detailed answers to questions about experiments:
>   - We beg to say the comment by the reviewer is a misinterpretation of what has been plotted in Figure 2(b). The plots do not show the SPS stepsize but a different statistic (SPS averaged both across clients and across the local steps). Therefore, it is possible that some client reaches the upper bound $\gamma_b$ in some local step---hence the difference in trajectories plotted in Figure 2(b).
>   - The purpose of Figure 2(b) was to show that adaptivity sets in from $\gamma_b = 1$, even if it is stable. Moreover, the purpose of Figure 2(a) was to show that all values of $\gamma_b$ leads to convergence of FedSPS, while that is not the case for FedAvg. So, our method is less sensitive to $\gamma_b$ than FedAvg is to $\gamma$.
>   - We are aware that in Figure 3(b), FedAMS shows a diverging behavior for non-i.i.d MNIST, but this is the plot obtained from our experiments. Note that the original paper on FedAMS [1] does not show any plot for non-i.i.d. data, and the original paper on FedADAM [2] has experiments for the EMNIST dataset and not MNIST. Nonetheless, we will verify this particular experiment regarding its correctness, for the potential camera-ready version.
>   - We wished to convey that our proposed method has lesser dependence on problem dependent parameters than previous adaptive federated methods, and its sensitivity to the parameters involved is lower. We agree with the reviewer in this regard, and shall rephrase Remark 4 to clarify this in the revised version of our manuscript.
> - (Q1) The term "adaptive methods" has a wide variety of connotations in the optimization literature: one of the earliest successful adaptive methods was AdaGrad, which is essentially just a stepsize. Adaptive methods can but do not necessarily have to involve gradient history and momentum. We chose our title to cater to the broader message that local adaptivity can be useful for Federated Learning.
> - (Q2) Our proof is based on error feedback analysis, and we can use similar ideas from followup work [3, 4] considering partial participation to extend our convergence analysis to partial client participation setting. However, this seems non-trivial and might be a direction for future work. We shall add a brief statement regarding this, as well as the suggested reference in our potential camera-ready version.
>
> **References:**
> - [1] Wang, Yujia, Lu Lin, and Jinghui Chen. "Communication-efficient adaptive federated learning." International Conference on Machine Learning. PMLR, 2022.
> - [2] Reddi, S. J., Charles, Z., Zaheer, M., Garrett, Z., Rush, K., Konecny, J., Kumar, S., and McMahan, H. B. Adaptive federated optimization. In International Conference on Learning Representations, 2021.
> - [3] Richtárik, Peter, Igor Sokolov, and Ilyas Fatkhullin. EF21: A new, simpler, theoretically better, and practically faster error feedback. In Advances in Neural Information Processing Systems, 2021.
> - [4] Fatkhullin, Ilyas, Igor Sokolov, Eduard Gorbunov, Zhize Li, and Peter Richtárik. EF21 with bells & whistles: Practical algorithmic extensions of modern error feedback. arXiv preprint arXiv:2110.03294, 2021.
>
> We believe that all the questions raised by reviewer vVZ6 have been satisfactorily answered in our response above. Overall, we should bear in mind to separately understand theoretical contributions and experimental insights. Many issues raised by the reviewer were quite interesting and thoughtful, and served to increase the standard of our manuscript, or can form the basis for interesting future work. We shall make sure to incorporate the clarifications and changes outlined above in the potential camera-ready version of our paper. If you agree that we managed to address all issues raised, please consider increasing your "Contribution" score, as well as the overall "Rating". If you believe this is not the case, please let us know so that we have a chance to respond further.

---

> > ### Comment · Reviewer_vVZ6 · 2023-11-23
> >
> > Thanks for the reply.
> >
> > Regarding Figure 2(b), there is no description of the what the curves are in the paper, so it may seem confusing. Please add a clarification, and/or consider adding a figure of the learning rate at individual clients to better demonstrate the benefit of adaptivity.
> >
> > For the experiments, I do think that there is an issue with the MNIST results, since it is basically the most standard dataset in the FL literature. It does not make much sense that FedAdam would perform well on EMNIST, FMNIST, CIFAR, etc., but diverge on MNIST.
> >
> > My question on the benefit of adaptivity in the experiments is resolved (please revise the figures and descriptions). Overall I think the quality of the paper is slightly above the threshold, and I will increase my score to 6. However, the issue in the experiments may suggest that the algorithms and baselines are not properly impelmented and compared.

---

### Official Review · Reviewer_g1co · 2023-11-03

**Soundness:** 3 good
**Presentation:** 3 good
**Contribution:** 2 fair
**Rating:** 5
**Confidence:** 4

**Summary:**

This work proposes a federated learning algorithm named FedSPS, and the proposed algorithm performs stochastic Polyak stepsize in local updates. Convergence is guaranteed under convex and strongly-convex cases. In particular, when the optimization objective is in the interpolation regime or the by choosing diminishing stepsize, exact convergence is guaranteed. Authors also provides various numerical evaluations of the proposed algorithm

**Strengths:**

1. The propose algorithm performs local adaptive gradient steps, in contrast, most existing adaptive gradient methods in FL perform adaptive gradients at the server side.
2. Theoretical analysis is provided. Approximate convergence for convex and strongly-convex cases are guaranteed and exact convergence is provided under two special cases: interpolation condition and small step-size condition.
3. Some numerical experiments are provided to validate the proposed algorithm. The numerical studies includes both ablation studies ($\gamma$, $c$, $\tau$ etc.) and comparison with baselines (FedAvg, FedAdam etc.)

**Weaknesses:**

1. The proposed algorithm seems to be a direct extension of Stochastic Polyak step to the federated learning setting.  What is the major difficulty of this application?
2. The theoretical analysis to the heterogeneity is not convincing. $\sigma_f^2$ is used as a measure of client heterogeneity in the paper, however, it is just an upper-bound (Proposition 1) of some more classical measure of heterogeneity, which means the proposed measure is weaker. In fact, if $l^*$ is chosen to be 0 (as in the paper), this measure is irrelevant to the heterogeneity.
3. Comparison with more baselines are desired. Although authors claim that "We design the first fully locally adaptive method for federated learning called FedSPS", there are already some local adaptive methods for FL, such as the  Local-AMSGrad method cited by the authors. It is desirable to add some comparison with these methods.

**Questions:**

Please see the weakness.

---

> ### Author Response · Authors · 2023-11-20
>
> We thank the reviewer for their evaluation of our manuscript, and provide detailed answers to the concerns below:
>
> - (W1) As explained in Remark 2, we experimented with various other algorithm design choices for incorporating SPS in FedAvg, such as FedSPS-Global and FedSPS-Normalized which are more complicated than our proposed FedSPS but did not offer any empirical benefits---hence our choice. Moreover, the primary theoretical challenge in analysis of FedSPS was extending the error-feedback framework (that originally works for equal stepsizes) to work for fully un-coordinated local stepsizes, and we did this for the first time in our work.
> - (W2) It is indeed true that the notion of optimal objective difference $\sigma_f^2$ is a weaker notion of heterogeneity than the standard ones in Federated Learning (FL). However, one can note that such a quantity appears naturally in the analysis of SPS-type methods, and the purpose of our Proposition 1 was to show that $\sigma_f^2$ is comparable to the standard heterogeneity assumptions in FL.
> - (W3) The existing algorithms for client-side adaptivity, such as Local-AdaAlter and Local-AMSGrad perform some form of aggregation of stepsizes during the communication round. Therefore, for these algorithms all the clients use the same stepsize for a particular round, and these methods are not fully locally adaptive as ours. Nonetheless, we agree with the reviewer that an empirical comparison with these methods would add value to the paper, and will include them in our potential camera-ready version.
>
> We believe that all the questions raised by reviewer g1co have been satisfactorily answered in our response above. We shall take care to incorporate the clarifications and changes outlined above in the potential camera-ready version of our paper. If you agree that we managed to address all issues raised, please consider increasing your "Contribution" score, as well as the overall "Rating''. If you believe this is not the case, please let us know so that we have a chance to respond further.

---

### Author Response · Authors · 2023-11-21
**Thanks to all reviewers and  general summary response**

We thank all the reviewers for carefully examining our manuscript and providing valuable suggestions. We tried to identify some common themes among all reviews, and provide some general comments below in response to the most important issues raised by the reviewers.

- **Novelty.** Our work is the first to show that locally adaptive federated optimization (i.e., using fully un-coordinated stepsizes on all clients) can be: (a) useful (ref. Example 1, where the optimal stepsizes $\gamma_1^{\star}$ and $\gamma_2^{\star}$ are calculated using the Polyak stepsize formula); and (b) theoretically prove its convergence (Thm. 3). Previous works either use constant stepsizes (FedAvg), or adaptivity only in the server side [1, 2].
- **Challenges.** The following challenges were involved in the design and analysis of the proposed method:
  - **Algorithm Design:** We arrived at the design choice for our proposed FedSPS algorithm by comparing its empirical performance to various other possible algorithmic choices such as FedSPS-Global and FedSPS-Normalized, both of which have higher complexity but comparable or worse performance than FedSPS (ref. Remark 2).
  - **Theoretical Analysis:** The proof techniques involved various technical challenges such as: (a) Extending the error-feedback framework (that originally works for equal stepsizes) to work for fully un-coordinated local (client) stepsizes; and (b) Extending the error-feedback framework (that originally works with bounded variance assumption) to work with the finite optimal objective difference assumption (that was used in the SPS paper).
- **Experiments.** We tried to clarify the various questions posed by reviewers regarding the experimental results. We shall add comparison to a few additional baselines (Local-AMSGrad and Local-AdaAlter) as requested by the reviewers in the potential camera-ready version of our paper.

**References:**
- [1] Wang, Yujia, Lu Lin, and Jinghui Chen. "Communication-efficient adaptive federated learning." In International Conference on Machine Learning, pp. 22802-22838. PMLR, 2022.
- [2] Reddi, Sashank, Zachary Charles, Manzil Zaheer, Zachary Garrett, Keith Rush, Jakub Konečný, Sanjiv Kumar, and H. Brendan McMahan. "Adaptive federated optimization." arXiv preprint arXiv:2003.00295 (2020).

---

### Meta-Review · Area_Chair_VcVN · 2023-12-07

**Metareview:**

Summary:
This work proposes a federated learning algorithm named FedSPS, and the proposed algorithm performs stochastic Polyak step size in local updates. Convergence is guaranteed under convex and strongly convex cases. In particular, when the optimization objective is in the interpolation regime or by choosing diminishing stepsize, exact convergence is guaranteed. The authors also provide various numerical evaluations of the proposed algorithm.

Strengths:
- The proposed algorithm performs local adaptive gradient steps, in contrast, most existing adaptive gradient methods in FL perform adaptive gradients at the server side.
- Simplicity of the algorithm.
- Theoretical analysis is provided. Approximate convergence for convex and strongly convex cases are guaranteed, and exact convergence is provided under two special cases: interpolation condition and small step-size condition.
- Some numerical experiments are provided to validate the proposed algorithm. The numerical studies include both ablation studies and comparisons with baselines (FedAvg, FedAdam, etc.).

Weaknesses:
- It was not clear what the difference is between stochastic polyak step size and this work (especially in practice).
- The theoretical analysis of the heterogeneity is not convincing; the authors agree that the notion is different from existing works.
- Comparison with more baselines is desired.
- The algorithm is not as adaptive as claimed.

**Justification For Why Not Higher Score:**

For the reasons above

**Justification For Why Not Lower Score:**

N/A

---

### Decision · Program_Chairs · 2024-01-16

Reject